# WINGS: Learning Multimodal LLMs without Text-only Forgetting

**Yi-Kai Zhang**[1,2,3*] **Shiyin Lu**[3] **Yang Li**[3] **Yanqing Ma**[3] **Qing-Guo Chen**[3]
**Zhao Xu**[3] **Weihua Luo**[3] **Kaifu Zhang**[3] **De-Chuan Zhan**[1,2] **Han-Jia Ye**[1,2†]
[1]School of Artificial Intelligence, Nanjing University
[2]National Key Laboratory for Novel Software Technology, Nanjing University
[3]Alibaba International Digital Commerce

## Abstract

Multimodal large language models (MLLMs), initiated with a trained LLM, first align images with text and then fine-tune on multimodal mixed inputs. However, during the continued training, the MLLM catastrophically forgets the text-only instructions that the initial LLM masters. In this paper, we present WINGS, a novel MLLM that excels in both text-only and multimodal instructions. By examining attention across layers of MLLM, we find that *text-only forgetting* is related to the attention shifts from pre-image to post-image text. From that, we construct an additional Low-Rank Residual Attention (LoRRA) block that acts as the "modality learner" to expand the learnable space and compensate for the attention shift. The complementary learners, like "wings" on either side, are connected in parallel to each layer's attention block. The LoRRA mirrors the structure of attention but utilizes low-rank connections to ensure efficiency. Initially, image and text inputs are aligned with visual learners operating alongside the main attention, balancing focus on visual elements. Later, textual learners are integrated with token-wise routing, blending the outputs of both modality learners collaboratively. Our experimental results demonstrate that WINGS outperforms equally-scaled MLLMs in both text-only and visual question-answering tasks. WINGS with *compensation of learners* addresses text-only forgetting during visual modality expansion in general MLLMs.

## 1 Introduction

Large Language Models (LLMs) [34, 54, 93, 115] are making significant strides toward Artificial General Intelligence (AGI) systems. Multimodal Large Language Models (MLLMs), as a visual expansion of LLMs, have demonstrated astonishing performance in vision-related captioning [14, 16, 68], understanding [7, 33, 122], and reasoning [117, 127, 133]. Common MLLMs build upon powerful pre-trained LLMs that take mixed textual and visual tokens as inputs. The visual ones are acquired using an image encoder and a projector. We describe instructions processed by the LLM without images as *text-only instructions*. In comparison, *multimodal instructions* incorporate visual feature tokens into text-only sequences. Modality fusing at the token level provides a flexible and effective pipeline for training MLLMs to comprehend visual information [77, 80, 81]. However, training on multimodal instructions seems to impair the pre-existing profound knowledge, especially making MLLM forget how to respond to text-only instructions like the initial LLM [86, 90]. MLLM experiences a drastic performance decline on text-only evaluation. We term it as the *text-only forgetting* of MLLM.

*Work done during the internship at Alibaba International Digital Commerce.
†Corresponding author, email: yehj@lamda.nju.edu.cn.

38th Conference on Neural Information Processing Systems (NeurIPS 2024).

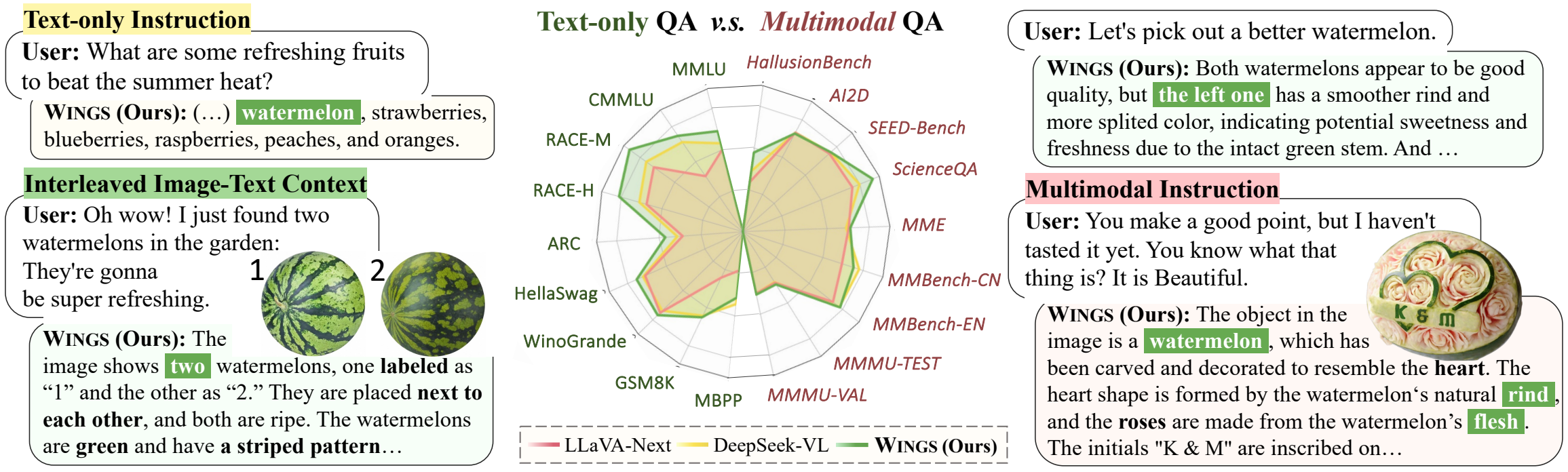

Figure 1: **Examples of text-only and multimodal conversations.** From left to right: Interacting with MLLM through *text-only* and *interleaved instructions*; Performance radar charts for WINGS, LLaVA-Next [81], and DeepSeek-VL [86] in *text-only* and *multimodal* QA tasks, with dark green indicating WINGS with the comprehensive performance; Interacting with *multimodal instructions*.

In practical applications, MLLMs also require engaging in text-only or interleaved conversations. As demonstrated in Figure 1, users often start with text-only inquiries and then, if not fully satisfied with the response, proceed to supplement questions with visual content. For multimodal instructions, MLLMs still rely on text to capture critical elements, as images may offer redundant information [15, 17, 85]. The first existing approaches replay extensive text-only or interleaved [61, 151] training data to mitigate catastrophic forgetting in MLLMs [72, 86, 90]. However, increasing training data incurs additional computational overhead and data collection challenges. Secondly, some applications [40] switch between LLM and MLLM based on whether images are included. This intuitive solution inevitably demands more deployment memory [1, 2] and is less cache-friendly in long vision-and-language interleaved conversations [42, 76, 101]. Therefore, it is crucial to train MLLM while preserving the text-only performance efficiently.

Given that the visual input tokens can be inserted at any position within the text sequence, we begin by examining the text before and after the inserted position to mark the impact of the visual part. Considering that MLLM's attention weights reflect the focus on tokens and influence the decision-making process, we first analyze the attention weights across each layer of the MLLM. Specifically, for each layer, we compute the attention weight proportion on all text tokens before and after the inserted image, termed as Layer-level Attention Weights (LAWS) of the before and after image text. From this, we examine the dynamic of attention across all layers as MLLM-Laws. Through training and sampling over 100 diverse MLLMs, we find that a well-trained model with superior text-only performance shows a positive correlation of MLLM-LAWS between the text segments before and after the image. This suggests that in a well-structured feature space, the main branch attention on text exhibits similar trends across layers, which is statistically linked to the semantic similarity of the text around the visual part. A closer similarity indicates minor disruption to MLLM's core attention, while a negative correlation shows that excessive focus on visual tokens shifts attention away from the text, significantly impacting MLLM-Laws.

Based on this observation, we propose WINGS, which introduces an extra module that acts as the boosted learner to compensate for the attention shift. We integrate complementary visual and textual learners in parallel at each layer's attention block, with visual learners enhancing focus on visual tokens and textual learners on text, respectively. In the first stage, visual features align with textual feature tokens, with all visual learners operating parallel to the main branch attention. The visual learners allocate some attention to visual tokens, mitigating the attention shift in the main branch. Subsequently, textual learners are integrated in parallel. We implement token-wise soft-routing based on shifted attention weights to harmonize the learning on visual and textual tokens. We design the Low-Rank Residual Attention (LoRRA) as the architecture for learners to ensure high efficiency. Figure 3 shows that the visual and textual learners on either side, like light feathers woven into "wings". Experiments show that our WINGS comprehensively achieves superior performance in text-only under the same training condition and exceeds other equal-level MLLMs on multimodal benchmarks. In addition, we construct the Interleaved Image-Text (IIT) benchmark with multi-turn

evaluations towards a general mixed-modality scenario. The samples are from text-only questions to strongly image-related conversations. WINGS achieve leading performance across various vision-relevance partitions. Overall, our contributions are as follows: (**1**) We claim and verify the text-only forgetting phenomenon of MLLM is related to the attention shift of cross-layer MLLM-LAWS before and after the image. (**2**) WINGS construct the visual and textual learners and introduce a router based on shifted attention weights for collaborative learning to compensate for attention shifts. (**3**) Experiments on text-only, visual-question-answering, and newly constructed Interleaved Image-Text (IIT) benchmarks demonstrate the comprehensive and versatile performance of WINGS.

## 2 A Closer Look at Attention Shift in Multimodal LLMs

In this section, we introduce the development from initialized LLM to MLLM. Next, we devise the MLLM-LAWS metric for representing attention shift and discuss the insights in building WINGS.

### 2.1 Granting Sight to Large Language Models

**Large Language Models (LLMs).** Even though existing Transformer-based [120] models [20, 83, 100, 134] like BERT [58] and OPT [142] have demonstrated profound language understanding capabilities, there has been a recent surge in powerful Generative Pre-trained Transformers (GPT) [10] under the auto-regressive language modeling paradigm. Both public [54, 55, 115, 116] and private [3, 93, 95, 112] solutions show remarkable progress in language comprehension and generation [91, 126]. These LLMs generally exceed a billion parameters, including pre-training [22, 32, 50, 56], supervised fine-tuning with instructions [26, 104, 110, 125], and reinforcement learning from human feedback [23, 96, 107, 152] on massive training data.

**Multimodal LLMs (MLLMs).** Integrating visual inputs into foundational LLMs to create MLLMs is becoming increasingly popular [18, 19, 63, 72, 136]. Unlike vision-centric multimodal frameworks [69, 137] such as CLIP series [99], MLLMs aim to align new modality features as the input of LLM with an additional encoder and perform multimodal question-answering [75, 80, 81, 128, 141, 150]. As illustrated in Figure 2 (a), it enables the combined training of mixed multimodal tokens, facilitating rapid deployment across various applications [24, 25, 45, 82, 122, 145]. One example of this pipeline is the LLaVA [80] series, which integrates a CLIP vision encoder with a linear projection to LLM Vicuna [21] and innovatively introduces instruction-following training data. Following this, some methods consider the richness of the vision-related training context [14, 46, 62], the scaled visual backbone [52, 73, 79], or the enhanced connectors [11, 124] to boost the visual effectiveness of MLLMs. Additionally, some works introduce monolithic multimodal solutions [30, 88, 111, 123]. Recently, some work has focused on the general capabilities of MLLM, specifically their performance on new modalities without suffering catastrophic forgetting of the text-only question-answering skills initially mastered by LLM [38, 74, 90]. For example, DeepSeek-VL [86] suggests that supplementing additional text-only training data can mitigate this forgetting. Others [78, 90, 114] try to incorporate interleaved visual-textual data into training to retain language knowledge. However, these methods are limited by training resources and data collection costs. We aim to preserve or even boost performance with text-related training data as little as possible. Some studies [66, 77, 106, 113, 135, 143] also consider expanding the scalability of LLM, such as using Mixture-of-Expert (MoE) with numerous parallel FFNs in the Transformer block alongside a sparse gating network for efficient selection [108, 148, 149]. There are some methods to configure effective information and feedback examples to enhance in-context learning abilities [131, 132]. These methods, however, require a massive increase in training parameters or inference costs. In WINGS, the newly designed parallel learners of Low-Rank Residual Attention (LoRRA) are similar to *MoE on attention block*, but with at least three orders of magnitude less in resource consumption. Compared to some LoRA-related methods [44, 47], WINGS focuses on parallel processing within the attention block rather than in certain in-block linear mappings [89, 121], particularly addressing the issue of capability forgetting in existing architectures.

### 2.2 Capturing the Attention Shift with MLLM-LAWS

The significant decline in text-only performance is closely linked to the observed related shift during the training process. Research on cross-modal learning [35, 67, 74] shows that transferring to new modalities affects feature distribution, output values, and activation levels. Considering attention

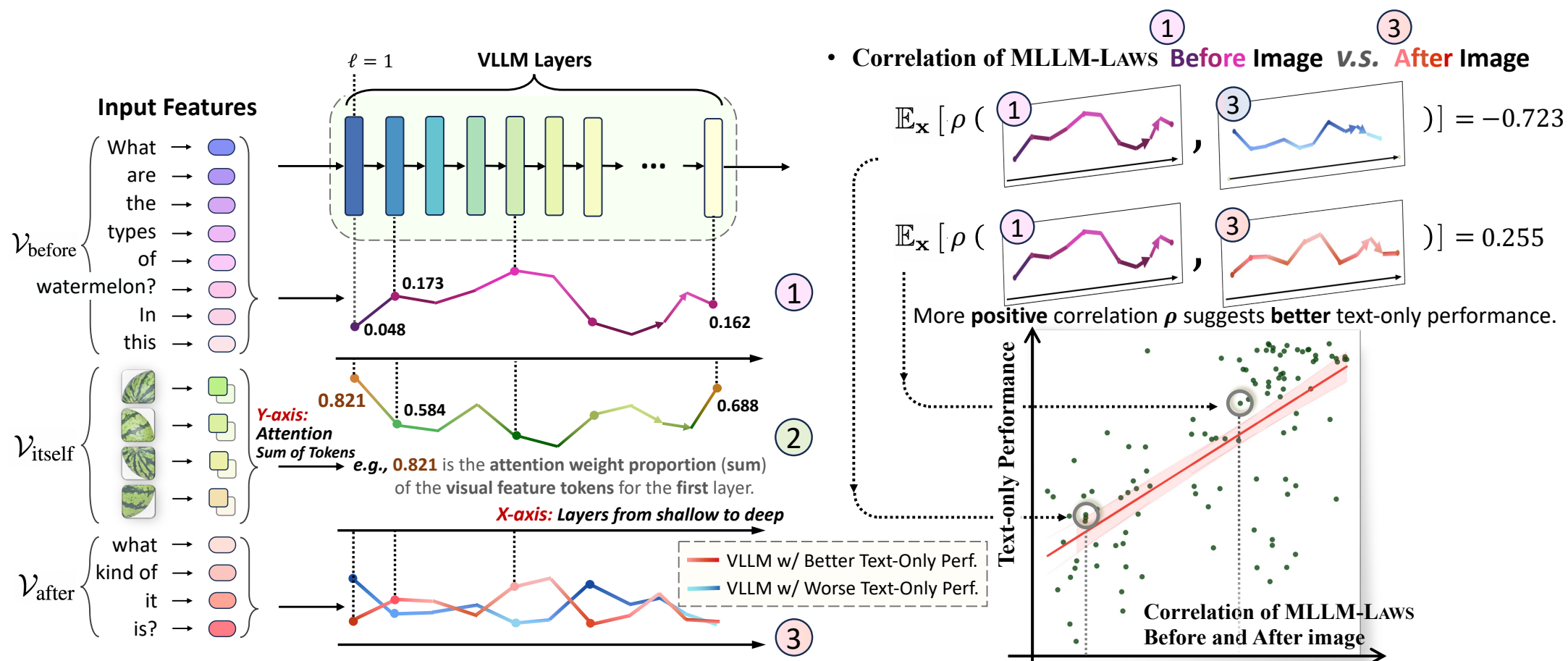

**(a)** The mixed visual and textual feature inputs. **(b)** Multimodal LLMs' Layer-level Attention Weights (MLLM-LAWS). **(c)** MLLM-LAWS Correlation Illustration

Figure 2: **Illustration of mixed visual-and-textual inputs and the Layer-level Attention Weights (LAWS) with its properties.** (a) The visual feature tokens from the visual encoder and projector are inserted into the textual feature sequence. (b) The attention weight proportion on textual tokens before-image, image-itself, and after-image across layers. The red curve is from the superior text-only MLLM, while the blue curve is from the inferior one. (c) Experiments on over 100 MLLMs show a positive correlation from the $\boldsymbol{\rho}$ for MLLM-LAWS before and after the visual tokens ($x$-axis) to the text-only performance of the MLLM ($y$-axis).

weights highlight where MLLM's focus depends on visual or textual tokens for decision-making [98], we investigate how attention shifts among *different parts of the sequences*, mainly where divided by the visual feature tokens. Specifically, we study over 100 diverse MLLMs to uncover how attention is allocated to each part for a text-only better MLLM. We take a closer look at the cross-layer dynamic curve of attention proportion on all text tokens *before* and *after* the inserted image.

For a instruction $\mathbf{x}$ and its hidden states in MLLM as $\mathbf{h} = [\mathbf{h}_1, \mathbf{h}_2, \cdots, \mathbf{h}_s]$ consisting of $s$ mixed visual and textual tokens. Let $\mathrm{a}_{ij}^{l}$ represent the attention weight between the $i^{\text{th}}$ and $j^{\text{th}}$ tokens in the $l^{\text{th}}$ of the $L$-layers MLLM. We have, for $\forall i$, $\sum_{j=0}^{s} \mathrm{a}_{ij}^{l}\left(\mathbf{h}^{l-1}\right) = 1$. As shown in Figure 2 (a), since the sequence of flattened visual tokens is continuously interleaved with the textual sequence, we denote the index set of the visual tokens as $\mathcal{V}_{\text{itself}} = \{v_{\text{start}}, v_{\text{start}} + 1, \cdots, v_{\text{end}}\}$. We refer to the textual sequence before the visual tokens as $\mathcal{V}_{\text{before}}$, and similarly, after the visual part as $\mathcal{V}_{\text{after}}$. For an MLLM with $L$ layers, we define the Layer-level Attention Weights (MLLM-LAWS) as:

$$\text{LAWS}_{\mathcal{V}_*} = \left[\mathrm{a}_{\mathcal{V}_*}^{1}, \mathrm{a}_{\mathcal{V}_*}^{2} \cdots, \mathrm{a}_{\mathcal{V}_*}^{L}\right], \quad \mathrm{a}_{\mathcal{V}_*}^{l} = \sum_{i=0}^{s} \sum_{j \in \mathcal{V}_*} \mathrm{a}_{ij}^{l}\left(\mathbf{h}^{l-1}\right), \tag{1}$$

where token index set $\mathcal{V}_*$ can be $\mathcal{V}_{\text{itself}}$, $\mathcal{V}_{\text{before}}$, or $\mathcal{V}_{\text{after}}$ as mentioned above, and for simplicity, we omit $\mathbf{h}^{l-1}$ in $\mathrm{a}_{\mathcal{V}_*}^{l}(\cdot)$ of $\text{LAWS}_{\mathcal{V}_*}$. In practice, $\text{LAWS}_{\mathcal{V}_*}$ characterizes the MLLM's attention on the current sequence $\mathcal{V}_{\text{itself}}$, $\mathcal{V}_{\text{before}}$, or $\mathcal{V}_{\text{after}}$ regarding the dynamic curve over all MLLM-layers. As shown in Figure 2 (b), the attention to the textual part initially increases and then decreases as the layers progress, while the trend for the visual one is often the opposite. We find that when the MLLM forgets the text-only instructions, the LAWS of the textual sequence after the visual ones show a deviation from the initial trend of rising and then declining. This implies a shift of layer-level attention in the text following the image $\mathcal{V}_{\text{after}}$ compared to that preceding the image $\mathcal{V}_{\text{before}}$. The dynamics labeled as ③ in Figure 2 (b) show the red curve for better text-only performance towards the worse blue one. To quantify this, we compute the Pearson Correlation Coefficient [92] between LAWS before and after the visual sequence. Formally,

$$\text{Attention Shift} = \mathbb{E}_{\mathbf{x}}\left[-\rho\left(\text{LAWS}_{\mathcal{V}_{\text{before}}}, \text{LAWS}_{\mathcal{V}_{\text{after}}}\right)\right] + 1 .$$

Studying the attention shift of over 100 diverse MLLMs, we find a positive correlation between the shift and the text-only performance degradation. In Figure 2 (c), each point represents a trained

MLLM. We find that one reason for the poor text-only performance of MLLMs is the misalignment of textual LAWS before and after the visual sequence, which largely stems from the main branch's attention block lacking sufficient capacity for continued fine-tuning [130, 147]. Next, We focus on how to mitigate the shifted attention weights. Starting with LAWS we give the MLLM "wings".

## 3 WINGS: Flying to Generality with Low-Rank Residual Attention Learners

In this section, we explore a sufficiently reliable and convenient mechanism to alleviate attention shifts. Specifically, we introduce an additional attention extension structure to assist in learning the main branch's attention. The WINGS architecture operates intuitively by incorporating visual and textual learners designed to mitigate shifts in attention. A dynamic attention-weighted router, guided by negative feedback from biased attention weights, adjusts the outputs of these visual and textual learners. WINGS aims to excel in text-only and visual question-answering tasks with high generality. We start with the typical training pipeline for MLLM like LLaVA [80] (subsection 3.1). Following this, we explore the motivation behind employing parallel modality learners and explain its implementation (subsection 3.2). Finally, we describe the training process for WINGS (subsection 3.3).

### 3.1 Revisit the Training Pipeline of the MLLM

Following the mainstream architecture of visual-encoder-based MLLM, we take mixed visual and textual features as inputs. For a one-turn conversation, the sequence of the visual feature tokens may appear at any position in the input $\mathbf{x}$. We represent the feature tokens as:

$$\mathbf{x} = [\mathbf{x}_{\mathtt{V}}, \mathbf{x}_{\mathtt{T}}] = \left[ \underbrace{\mathbf{h}_1, \cdots,}_{\text{textual features}} \underbrace{\mathbf{h}_{v_{\text{start}}}, \mathbf{h}_{v_{\text{start}}+1}, \cdots, \mathbf{h}_{v_{\text{end}}},}_{\text{visual features}} \underbrace{\cdots, \mathbf{h}_s}_{\text{textual features}} \right] , \tag{2}$$

where we omit the superscript of layer-index $l$ for the $0^{\text{th}}$ layer. The $v_{\text{start}}$ and $v_{\text{end}}$ represent the starting and ending indices of the visual feature tokens, usually obtained through the vision encoder $\boldsymbol{\psi}$ and projector $\mathbf{W}_{\text{proj}}$, as $\mathbf{x}_{\mathtt{V}} = \mathbf{W}_{\text{proj}} \cdot \boldsymbol{\psi}\left(\mathbf{x}_{\text{image}}\right)$. Correspondingly, $\mathbf{x}_{\mathtt{T}} =$ the remaining $0$ to $v_{\text{start}}$ and $v_{\text{end}}$ to length $s$ denote features of the textual system prompt or user instructions. We consider the posterior of the ground-truth answer as:

$$\Pr\left(\mathbf{x}_{\mathtt{a}} \mid \mathbf{x}\right) = \prod_{i=1}^{s} \mathbb{1}_{[1, v_{\text{start}}) \cup (v_{\text{end}}, s]} \cdot \boldsymbol{\varphi}\left(\mathbf{h}_i \mid [\mathbf{h}_1, \cdots, \mathbf{h}_{i-1}]\right) . \tag{3}$$

Here, $\boldsymbol{\varphi}$ represents the main branch LLM, which consists of Transformer decoder layers [119]. Considering the interleaved image tokens, we omit the loss calculation for the "next visual token."

### 3.2 Visual and Textual Learners Weave WINGS

**Motivation: Learning to mitigate the attention shift with modality-specific auxiliary structures.** As mentioned in subsection 2.2, MLLM-Laws demonstrates the attention shift in the sequence following the visual features. The shift results from excessive dependency on visual features. This issue may stem from the insufficient alignment within mixed inputs [7, 15, 130], or the main branch struggles to accommodate capacity expansion during continual learning, where new modalities can obscure existing knowledge. It suggests adding a small, adjustable factor to the shifted mixed modality features and regulating unnecessary fluctuations in MLLM-LAWS. Consequently, we aim to adopt an efficient, learnable module as the visual "wing". Compared to the image-text mixed feature inputs of the main branch, it should focus specifically on extracting visual information to share the burden of overly shifted attention. The interaction between the current hidden state and visual features is conducted within this module. Similarly, to balance the auxiliary function of the visual learner, we also construct a symmetrical textual learner. Moreover, it is crucial to appropriately distribute the two learners across both modalities to ensure they function collaboratively.

**Structure: parallel learner of attention & token-wise router of attention outputs.** To capture key information in shifted modalities while ensuring efficiency, we design a multihead Low-Rank Residual Attention (LoRRA) learner at every layer. It takes input from the hidden state and interacts with the initial visual or text-only feature. The learner facilitates cross-cascading with the initial

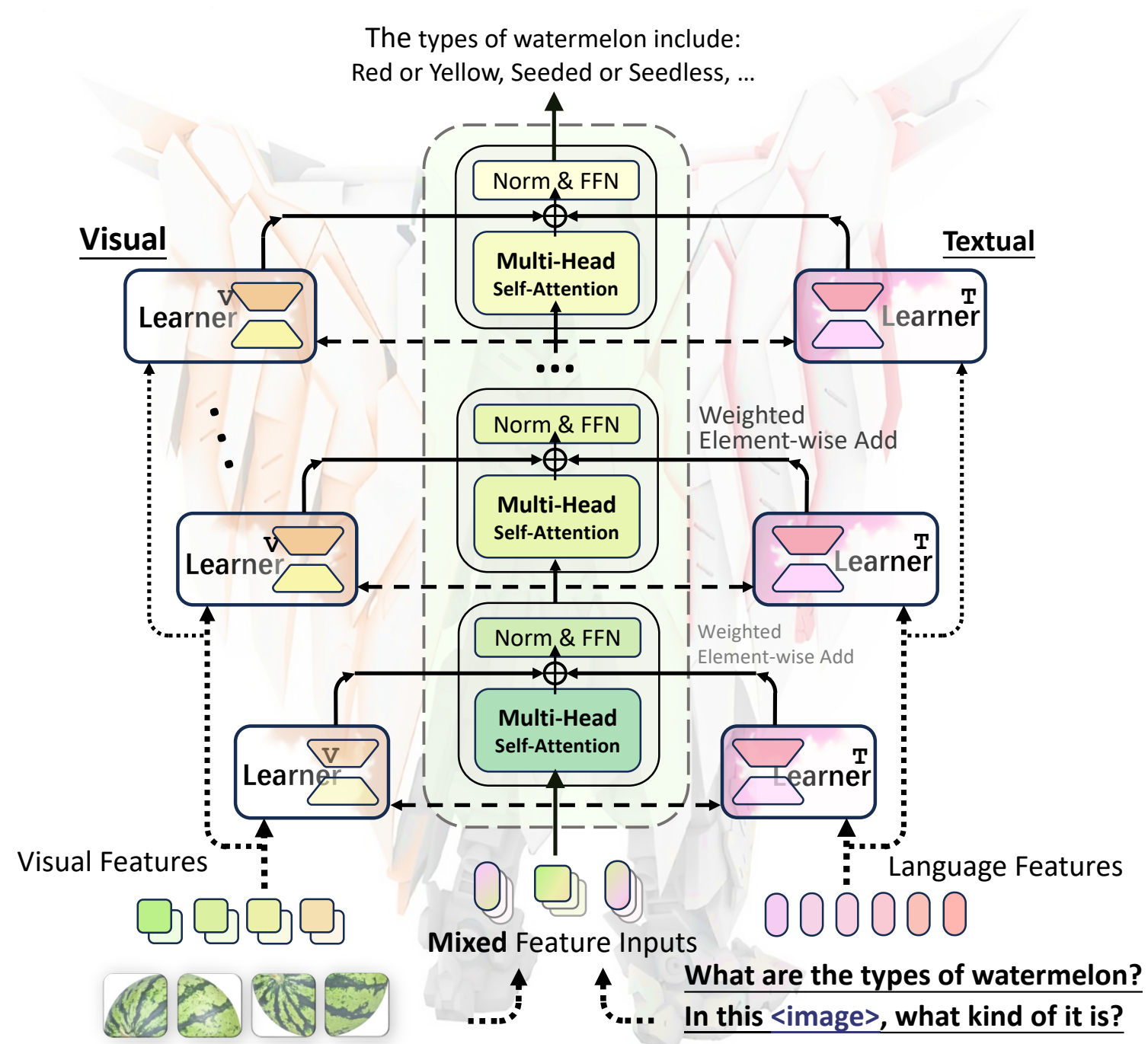

Figure 3: **The WINGS - model architecture.** We introduce extra modules parallel to the main attention, serving as boosted learners to compensate for the attention shift. We train the visual learners on one side, alleviating some shifted attention. Then, we collaboratively learn visual and textual learners based on routing shifted attention weights. They are like light feathers woven "wings".

projected information. Specifically, for the $l^{\text{th}}$ layer, the visual/text-only learner is formulated as:

$$\text{Learner}^{*}\left(\text{Q=}\mathbf{h}^{l},\text{K,V=}\mathbf{x}_{*}\right)_{*\in\{\texttt{V},\texttt{T}\}}=\text{Softmax}\left(\frac{\mathbf{h}^{l}\left(\mathbf{1}+\mathbf{W}^{\mathsf{Q}}\right)\cdot\left(\mathbf{x}_{*}\left(\mathbf{1}+\mathbf{W}^{\mathsf{K}}\right)\right)^{\top}}{\sqrt{d_{\text{head}}}}\right)\mathbf{x}_{*}\left(\mathbf{1}+\mathbf{W}^{\mathsf{V}}\right)\mathbf{W}^{\mathsf{O}}\,, \tag{4}$$

where the matrix $\mathbf{W}^{\mathsf{Q}}$, $\mathbf{W}^{\mathsf{K}}$, $\mathbf{W}^{\mathsf{V}}$, and $\mathbf{W}^{\mathsf{O}}$ is low-rank and is obtained by the dot product of $\mathbf{W}_a \in \mathbb{R}^{d\times \underline{d}}$ and $\mathbf{W}_b \in \mathbb{R}^{\underline{d}\times d}$, and $\underline{d}$ is relatively small enough. The symbol $\mathbf{1}$ is represented as the identity matrix. The structure of multihead LoRRA preserves the effectiveness of the cross-attention structure and employs efficient low-rank mapping to reduce computational demands. Following LoRA [47], LoRRA learners also employ random Gaussian initialization for $\mathbf{W}_a$ and sets $\mathbf{W}_b$ to zero. Since $\mathbf{W}^{\mathsf{O}}$ lacks a residual, the output of LoRRA is zero at the beginning of training. As shown in Figure 3, the visual and textual features are fed into their respective side learners, like two "wings" woven together. The outputs of two learners from each layer are then weighted sum to the attention of the main branch. As illustrated in the left part of Figure 4, a router receives attention weights to generate the balance weights of visual and textual learners for each token. In summary, we formulate the WINGS block as:

$$\text{Att}^{\text{WINGS}}=\text{Att}^{\text{main}}+\sum_{*\in\{\texttt{V},\texttt{T}\}}\text{Router}\left(\mathbf{a}\right)\cdot\text{Learner}^{*}\left(\mathbf{h}^{l},\mathbf{x}_{*}\right)\,, \tag{5}$$

where $\mathbf{a} \in \mathbb{R}^{s\times s}$ represents the attention weights of the current main branch. The router is formalized as $\text{Router}\left(\mathbf{a}\right)=\mathbf{W}[:,:s]\cdot\mathbf{a}^{\top}$, which is implemented by a single-layer dynamic MLP, $\mathbf{W} \in \mathbb{R}^{2\times s_{\max}}$. It receives the attention weights $\mathbf{a}$ and processes them using $\mathrm{Softmax}$ on two modality learners.

### 3.3 Stable Training Recipe

The architecture of WINGS comprises four elements: vision encoder, projector, initialized LLM, and the learners with a router. During the training process, the vision encoder is consistently fixed. Firstly,

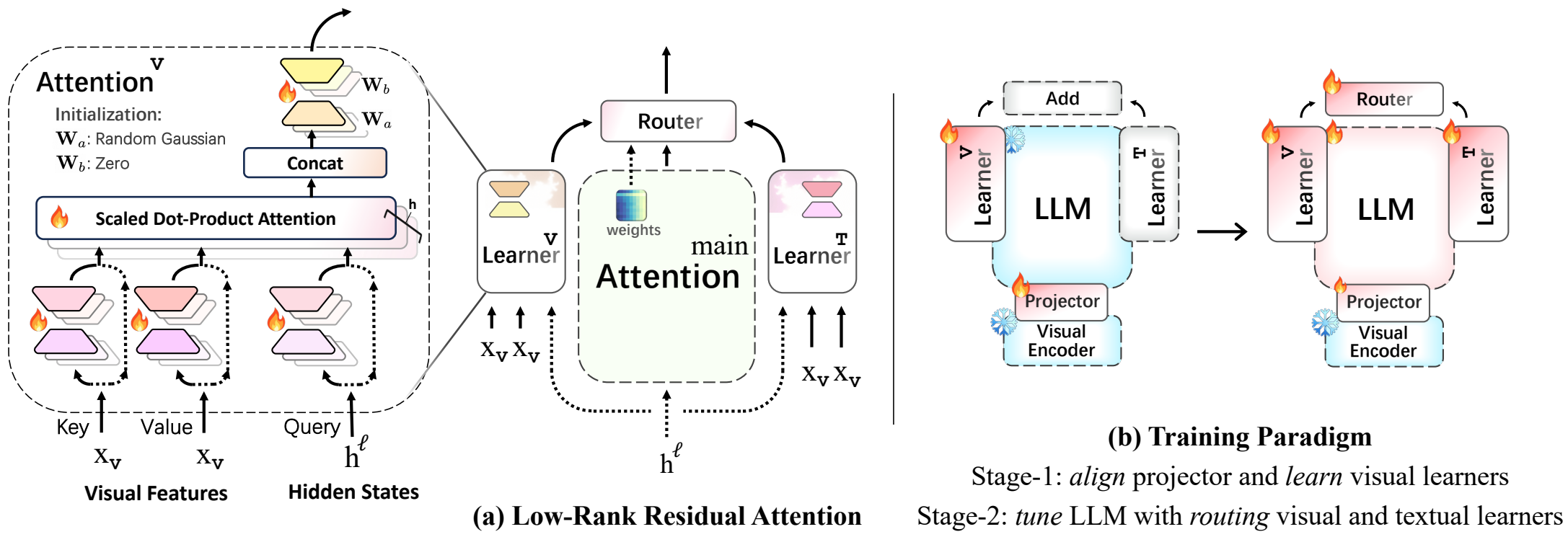

Figure 4: **Illustrations of the detailed WINGS structure, and training strategies.** WINGS is constructed by the Low-Rank Residual Attention (LoRRA) module where the previous hidden state acts as the `query` and the visual/textual features serve as the `key` and `value`. Training starts with visual learners and projectors, followed by the dynamic attention-based routing.

we only fine-tune the projector and visual learners. We primarily employ image-text pairs for visual alignment, while the outputs of visual learners are directly added to the main branch. For this part, the visual learners primarily handle the visual focus, minimizing disturbances to the main branch during continued learning. Subsequently, textual learners are paralleled with visual learners on the attention block of LLMs. The router begins by learning to allocate visual and textual learners from the attention weights of the main branch. At this stage, both types of learners work more effectively together to focus attention on key tokens. To summarize, WINGS prioritizes enhancing visual learners first. Subsequently, it "spreads its wings" by concurrently learning and routing visual and textual learners based on shifted attention weights. During inference, the routed weights of the visual wings branch are deactivated for text-only instructions, while multimodal instructions activate both wings.

## 4 Experiments

In this section, we first introduce the benchmarks for evaluating WINGS, including Table 1: text-only forgetting on the same multimodal training data, Table 2: comparison with general MLLMs, and Figure 5: analysis on the Interleaved Image-Text (IIT) benchmark with varying levels of vision-related conversation. Following that, we outline the training details and configurations of the WINGS, and delve into experimental analysis across each benchmark. Following that, we perform an ablation study on various learning rates with different training parts. Finally, we provide supplementary descriptions regarding WINGS' overhead compared to general MLLMs and how its innovative compensatory learners help effectively mitigate attention issues.

**Evaluation Setups.** We aim to assess through MLLM how much visual information is required for evaluation. For example, generic multimodal instructions require MLLMs to strongly capture image aspects, whereas text-only instructions focus on the text. We introduce three types of benchmarks:

- **Standard text-only benchmarks.** We are particularly interested in the text-only performance improvement of WINGS under the same training data and resource conditions. Different datasets including *interdisciplinary exams* like MMLU [43], CMMLU [65], ARC-Easy, ARC-Challenge [27], language *understanding* and *knowledge* such as WinoGrande [103], Openbook QA [8], Race-Middle, Race-High [60], WSC [129], CHID [144], *reasoning* such as HellaSwag [139], SIQA [105], PIQA [9], OCNLI [48], and *math* and *code*-related tasks such as GSM8K [28] and MBPP [4] are comprehensively evaluated.
- **General multimodal benchmarks.** We evaluate on MMMU [138], MME [37], MMBench [84] (MMB) in English (EN) and Chinese (CN), ScienceQA [87] for test (SciQA), SEED-Bench [64] for image part (SEED), AI2D [57] for test, and HallusionBench [41] (HallB).
- **Our Interleaved Image-Text (IIT) benchmark** with diverse text-only, interleaved, and image-ralated multi-turn conversations. It includes sampling for MMLU, CMMLU, OpenbookQA, HellaSwag, MMMU, MMBench, SEED-Bench, and AI2D datasets.

| Dataset | | Vicuna LLM | Vicuna + CLIP | LoRA$_{\text{Vicu.}}$ + CLIP | Vicuna + SigLIP | Qwen LLM | Qwen + CLIP | LoRA$_{\text{Q}}$ + CLIP | Qwen + SigLIP | WINGS (Ours) | Text-only Forgetting ( - ) | Our Impro. ( - ) |
|---|---|---|---|---|---|---|---|---|---|---|---|---|
| Exam | MMLU | 51.18 | 51.12 | 48.89 | 50.63 | **60.86** | 50.83 | 59.67 | 51.16 | 60.53 | 9.70 | 9.37 |
| | CMMLU | 38.60 | 38.29 | 37.24 | 38.73 | 69.37 | 62.58 | 67.87 | 60.46 | **69.82** | 8.91 | 9.36 |
| | ARC-E | 57.62 | 53.63 | 55.82 | 53.95 | **59.96** | 56.93 | 59.35 | 55.87 | 54.29 | 4.09 | -1.58 |
| | ARC-C | 33.75 | 34.60 | 34.68 | 35.17 | 38.90 | 39.14 | 38.64 | 39.50 | **43.39** | -0.60 | 3.89 |
| Under-standing | Winogrande | 68.01 | 64.97 | 67.83 | 65.21 | **71.38** | 69.82 | 71.03 | 69.05 | 69.28 | 2.33 | 0.23 |
| | OpenbookQA | 77.10 | 73.28 | 77.15 | 72.12 | **81.73** | 78.31 | 81.29 | 77.51 | 81.05 | 4.22 | 3.54 |
| | Race-Middle | 63.99 | 60.10 | 62.84 | 59.45 | **74.82** | 68.25 | 72.06 | 68.34 | 74.24 | 6.48 | 5.90 |
| | Race-High | 58.74 | 53.24 | 54.91 | 52.69 | **71.05** | 59.20 | 65.67 | 57.72 | 69.62 | 13.33 | 11.90 |
| | WSC | 51.30 | 47.21 | 51.06 | 47.72 | 56.17 | 54.18 | 57.30 | 55.23 | **66.35** | 0.94 | 11.12 |
| | CHID | 39.05 | 49.66 | 45.26 | 53.49 | 71.94 | 71.82 | 72.92 | **74.29** | 74.06 | -2.35 | -0.23 |
| Reasoning | HellaSwag | 63.11 | 63.08 | 62.58 | 63.02 | **65.70** | 61.90 | 64.32 | 63.24 | 65.12 | 2.46 | 1.88 |
| | SIQA | 42.37 | 44.06 | 43.27 | 44.52 | 45.57 | 50.20 | 46.83 | **51.71** | 49.64 | -6.14 | -2.07 |
| | PIQA | 71.92 | 71.95 | 70.35 | 71.84 | 76.59 | 74.60 | 73.77 | 75.19 | **78.06** | 1.40 | 2.87 |
| | OCNLI | 33.89 | 37.74 | 39.41 | 40.46 | 49.73 | 48.31 | 48.07 | 50.29 | **50.39** | -0.56 | 0.10 |
| Math | GSM8K | 25.19 | 23.72 | 22.68 | 23.05 | 56.77 | 50.10 | 54.25 | 51.37 | **52.08** | 5.40 | 0.71 |
| Code | MBPP | 13.80 | 11.29 | 13.92 | 10.80 | 37.50 | 34.82 | 36.72 | 33.20 | **38.92** | 4.30 | 5.72 |
| Multimodal | MMMU-VAL | – | 35.67 | 30.78 | 35.56 | – | 34.56 | 32.33 | 35.11 | **39.89** | – | 4.78 |
| | MMMU-TEST | – | 34.40 | 30.90 | 35.33 | – | 34.90 | 31.80 | 35.10 | **37.30** | – | 2.20 |
| | MMBench | – | 63.18 | 59.83 | 65.14 | – | 66.05 | 62.84 | **70.94** | 70.53 | – | -0.41 |
| | ScienceQA | – | 67.72 | 64.49 | 71.50 | – | 74.26 | 69.09 | 74.89 | **78.76** | – | 3.87 |

Table 1: **Performance comparisons of WINGS and the baseline MLLMs under the same training data**. We consider 8 baseline MLLMs, including LLMs as Vicuna$_{\text{v1.5}}$ & Qwen1.5, visual encoders as CLIP [99] & SigLIP [140], and training strategies as full-parameter & LoRA fine-tuning. The first entry represents the initial LLM, upon which each MLLM is trained. Our evaluation spans 6 domains with 20 datasets. WINGS is based on the Qwen1.5 and SigLIP, and the column "Our Improvement" highlights how much WINGS surpasses its baseline with the same backbones.

**Model Summaries & Implementation Details.** We release the WINGS$_{\texttt{base}}$ and WINGS$_{\texttt{pro}}$, with Qwen1.5-7B LLM [6] and SigLIP [140] visual encoder as the foundations. We also introduce the WINGS$_{\text{1.8B}}$ version, adapted to Qwen1.5-1.8B LLM for edge device compatibility. As illustrated in Figure 4, we only optimize the projector and the image learners of WINGS for the first alignment stage. The LLM branch adaptation is incorporated during the second instruction tuning stage. We train for 1 epoch with the AdamW optimizer and the Cosine learning schedule. Typically, the learning rates for the first and second stages are set at $1e^{-3}$ and $2e^{-6}$ (with the projector part as $1e^{-5}$), respectively. For WINGS$_{\texttt{base}}$, approximately 1m training data to align image learners and about 0.6m supervised fine-tuning instructions for the next stage (the same as LLaVA$_{\text{v1.5}}$ [80]). In the WINGS$_{\texttt{pro}}$, we use the same aligned data and approximately 2m training data for learning image-text learners. These two types of MLLM require about 1.5 and 6 days of training on 8$\times$ A100 GPUs, respectively. The training datasets for WINGS$_{\texttt{mini}}$ are consistent with the WINGS$_{\texttt{pro}}$. It takes approximately 5 days to run on 4$\times$ A100 GPUs.

**Details in Figure 2.** We adopt various multimodal to text-only sample ratios (`25:1`, `20:1`, `10:1`, `5:1`, `2:1`, `1:1`, `1:2`, . . ., `1:25`) plus an `all:0` setup (12 combinations total) to ensure sufficient scenarios. The learning rate is kept consistent with the setup described above. We sample 5 models per epoch, excluding 12 failed ones due to issues like gradient explosion, resulting in 108 for analysis.

### 4.1 Toward Comprehensive Text-only and Multimodal Performance

**Text-only Comparison in Fair Data and Resource Environments.** As shown in Table 1, "Vicuna-v1.5 + CLIP" corresponds to LLaVA$_{\text{v1.5}}$, and "Qwen1.5 + SigLIP" serves as the foundation for WINGS. When comparing LLM itself and the rest of MLLMs, we observe that fine-tuning with multimodal instructions, compared to the "Qwen LLM", there is text-only forgetting in 12 out of 16 datasets, with notable decreases of up to 9.70, 8.91, and 13.33 in MMLU, CMMLU, and RACE-High, respectively. WINGS significantly improve performance on datasets such as MMLU, CMMLU, RACE-High, and WSC, despite the potential for severe text-only forgetting on baselines. Additionally, we find that

| Method \ Dataset | Text-Only QAs | | | | | | | Multimodal QAs | | | | | | |
|---|---|---|---|---|---|---|---|---|---|---|---|---|---|---|
| | MMLU/C* | RACE-M/H | ARC | HellaSwag | Winog. | GSM8K | MBPP | MMMU-V/T | MMB-EN/CN | MME | SciQA | SEED | AI2D | HallB |
| *Equal-Scale Open-Source 7B Multimodal LLMs* | | | | | | | | | | | | | | |
| O-Flamingo$_{v2}$ [5] | 26.9 / 27.1 | 40.3 / 32.6 | 31.0 | 55.4 | 58.3 | 10.2 | 9.1 | 29.1 / 28.7 | 10.9 / 13.3 | 803.9 | 55.8 | 30.2 | 32.6 | 30.4 |
| IDEFICS [51] | 33.0 / 26.4 | 38.2 / 36.9 | 33.2 | 58.9 | 60.2 | 11.7 | 8.1 | 17.6 / 20.2 | 49.6 / 27.3 | 1239.3 | 62.4 | 44.8 | 43.4 | 24.6 |
| InstructBLIP [29] | 43.2 / 35.7 | 52.8 / 49.7 | 39.5 | 55.7 | 54.9 | 18.3 | 10.3 | 32.7 / 32.1 | 38.5 / 26.8 | 1425.6 | 61.3 | 45.7 | 41.1 | 33.3 |
| ShareGPT4V [14] | 47.6 / 36.9 | 55.9 / 51.0 | 41.6 | 54.7 | 60.1 | 18.0 | 8.9 | 35.5 / 35.2 | 67.4 / 63.1 | **1915.3** | 68.9 | 68.1 | 58.2 | 26.6 |
| Qwen-VL [7] | 49.7 / 58.3 | 65.2 / 64.8 | 34.4 | 58.2 | 61.0 | 49.0 | 34.6 | 36.4 / 35.9 | 60.3 / 57.4 | 1806.2 | 69.6 | 62.0 | 61.9 | 34.1 |
| Monkey [73] | 52.8 / 66.9 | 65.6 / 62.1 | 38.2 | 60.6 | 59.3 | 51.8 | 37.1 | **40.3** / 37.1 | 71.9 / 67.8 | 1815.4 | 78.3 | 69.1 | 62.5 | 42.1 |
| LLaVA$_{v1.5}$ [80] | 51.1 / 38.3 | 60.1 / 53.2 | 34.6 | 63.1 | 65.0 | 23.7 | 11.3 | 35.7 / 34.4 | 63.2 / 57.7 | 1518.6 | 67.7 | 63.7 | 56.4 | 29.7 |
| LLaVA$_{Next}$ [81] | 50.2 / 39.7 | 65.1 / 58.3 | 36.0 | 63.7 | 68.9 | 30.3 | 23.0 | 37.6 / 35.8 | 67.8 / 61.8 | 1760.3 | 70.1 | 69.1 | 66.4 | 29.6 |
| DeepSeek-VL [86] | 53.9 / 64.0 | 70.6 / 63.8 | 39.2 | 65.1 | 67.2 | 55.3 | **43.1** | 37.6 / 35.3 | 72.7 / **72.5** | 1716.8 | 80.6 | 70.0 | **66.5** | 36.2 |
| **WINGS** (Ours) | 60.5 / **69.8** | 74.2 / 69.6 | 43.4 | 65.1 | 69.3 | 52.1 | 38.9 | 39.9 / **37.3** | 70.5 / 68.3 | 1753.8 | 78.8 | 69.5 | 62.7 | 45.8 |
| **WINGS$_{pro}$**(Ours) | **61.3** / 68.5 | **82.8 / 76.3** | **46.3** | **69.2** | **70.9** | **56.3** | 39.3 | 38.2 / 36.9 | **73.1** / 69.0 | 1786.1 | **83.1** | **70.2** | 65.8 | **47.3** |
| *Advanced Private Multimodal LLMs* | | | | | | | | | | | | | | |
| GPT-4 [95] | 83.5 / 71.2 | 93.2 / 87.8 | 93.6 | 88.4 | 75.6 | 91.6 | 56.2† | – | – | – | – | – | – | – |
| GPT-4V [94] | 79.3 / 69.4 | 93.7 / 89.2 | 92.9 | 84.7 | 76.1 | 88.4 | 72.4 | 58.9 / 56.8 | 77.0 / 74.4 | 2153.6 | 68.4 | 73.7 | 75.5 | 46.5 |
| Gemini$_{pro\ vision}$ [102] | 85.9 / 73.7 | 88.9 / 83.2 | 85.0 | 78.8 | 71.5 | 86.4 | 61.5 | 60.6 / 62.2 | 73.6 / 74.3 | 2193.2 | 58.3 | 70.8 | 70.2 | 45.2 |
| *Efficient Multimodal LLMs with* WINGS$_{1.8B}$ | | | | | | | | | | | | | | |
| DeepSeek-VL$_{1.3B}$ [86] | 31.7 / 38.2 | 63.6 / 58.4 | 35.8 | **52.9** | 45.7 | 17.6 | 16.3 | 33.8 / 32.3 | 65.1 / 60.7 | 1483.4 | 65.4 | 63.3 | 50.1 | 25.0 |
| MiniCPM-V$_{2.4B}$ [49] | 42.4 / 40.9 | **68.8** / 62.6 | 37.0 | 48.3 | 51.7 | 32.5 | 24.2 | **37.2 / 34.4** | **65.7 / 64.1** | **1584.1** | 64.9 | **64.7** | 54.9 | **31.8** |
| **WINGS$_{1.8B}$**(Ours) | **44.9 / 50.9** | 68.5 / **63.2** | **37.1** | 50.5 | **53.0** | **40.6** | **28.5** | 35.7 / 33.9 | 64.2 / 61.2 | 1527.3 | **67.5** | 62.8 | **55.2** | 30.2 |

Table 2: **Performance comparisons of the equal-scale MLLMs and the efficient multimodal LLMs** on text-only and multimodal datasets. We evaluate the open-source, efficient, and private API MLLMs. We select 18 representative evaluation datasets. C* represents the CMMLU dataset.

the forgetting effects of CLIP and SigLIP are similar. In contrast, parameter-efficient fine-tuning methods like LoRA result in less text-only forgetting but underperform on multimodal questions. Overall, WINGS' visual and textual learners are credibly demonstrated to retain performance on text-only tasks while also performing well on visual-related questions. In datasets like CHID, OCNLI, and SIQA, MLLMs show improved text-only performance due to increased language diversity (*e.g.*, Chinese context) or semantic similarity in their fine-tuning data.

**General Evaluation in Text-Only and Multimodal Tasks.** We present the performance of 9, roughly 8B open-source MLLMs, 2 roughly 2B, and 2 private API ones evaluated in the general text-only and multimodal tasks. Table 2 shows that WINGS series can perform better on text-only and multimodal question-answering datasets. It achieves state-of-the-art performance on 13 out of 18 datasets, significantly surpassing LLaVA$_{v1.5}$ with the same architecture. We find that WINGS is equally effective for more efficient foundations, as shown in the "Efficient Multimodal LLMs" parts. WINGS can still capture key elements and demonstrate good scalability as the parameter increases. Although WINGS$_{\texttt{base}}$ does not receive additional training for the text-only component, it is still able to achieve comparable performance.

### 4.2 Interleaved Image-Text (IIT) Benchmark

To finely evaluate MLLMs, we construct a series of text-only and multimodal mixed multi-turn conversations. We extract instructions from MMLU, CMMLU, OpenbookQA, HellaSwag, MMMU, MMBench, SEED-Bench, and AI2D datasets with similar semantics by chroma [39]. We then polish the connection between some instructions using GPT-3.5 Turbo to make them closer to real-world conversations. We set up 6 vision-content configurations, categorized by the multi-turn content as: (`T`), (`T`, `T`), (`T`, `T`, `T`), (`T`, `T`, `V`), (`T`, `V`), and (`V`). For instance, (`T`, `T`, `V`) indicates two consecutive text-only queries followed by a visual question requiring a response.

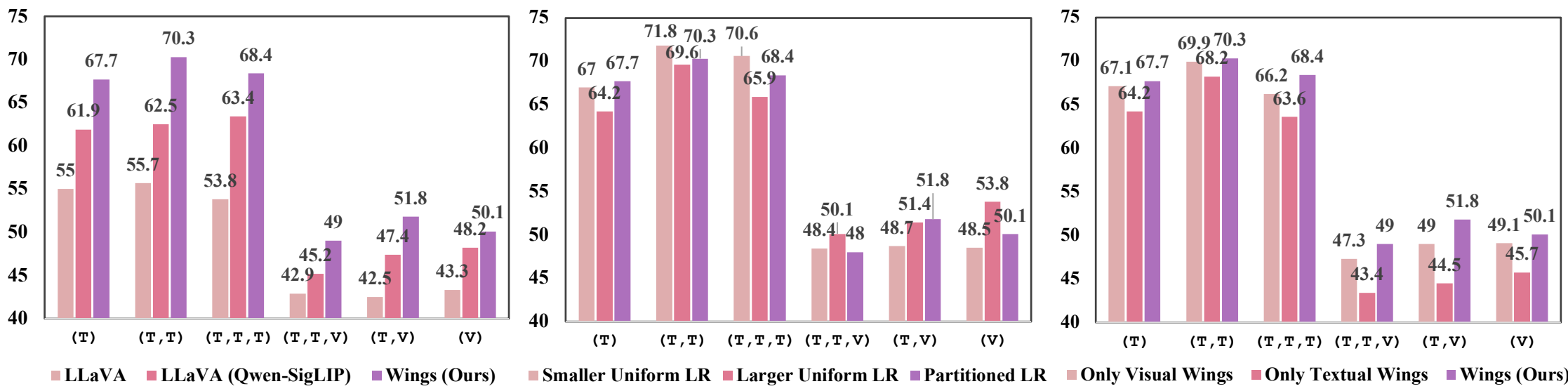

**(a)** Comparison of LLaVA-v1.5 Series and WINGS **(b)** Comparison of Various Learning Rate on WINGS **(c)** Comparison of Different Training Parts of WINGS

Figure 5: **Performance comparison** on the newly constructed **Interleaved Image and Text (IIT) Benchmark** of the **LLaVA series**, **different learning rate** and **fine-tuning parts**. The horizontal axis represents different multimodal question settings. The horizontal axis shows different multimodal setups, *e.g.*, (T, T, I) represents a visual question after two text-only QAs. The three subfigures represent different ablation settings, with the violet color representing our WINGS.

### 4.3 Ablation Studies

Referencing Figure 5, we address three questions to comprehensively analyse WINGS:

- Can WINGS sustain performance with interleaved evaluation? We find that part (a) highlights WINGS surpassing $\text{LLaVA}_{\text{v1.5}}$ and the same-backbone as $\text{LLaVA}_{\text{v1.5}}$ (Qwen-SigLIP) for each multi-turn setting, especially in text-centric dialogues.
- How do WINGS fare with different learning rate settings? Part (b) demonstrates that using a lower learning rate maintains proficiency in text-only tasks but falls short in multimodal questions, while a higher rate boosts multimodal abilities but not text-only. Applying a higher learning rate to the projector and a lower one to the others achieves the optimal.
- Are all components of WINGS equally effective? In part (c), we examine that incorporating visual learners alone slightly preserves text-only abilities, likely by minimizing disruption to the LLM, but diminishes performance on multimodal tasks.

In the diverse IIT bench, which ranges from text-rich to multimodal contexts, the effectiveness of WINGS is particularly evident. As shown in Figure 1, within real-world applications, textual content offers insights for following visual tasks. WINGS excels in handling text-only tasks while improving performance on visual-related instructions.

## 5 Conclusion

We propose WINGS, which includes visual and textual learners, to alleviate text-only forgetting. The learner is composed of efficient Low-Rank Residual Attention (LoRRA). We start by considering the shifted attention weights in MLLM and, in the first stage, focus on learning the visual learner. Then, we co-train the visual and textual learners with routing based on the shifted attention weights. WINGS demonstrates remarkable performance on text-only, visual-question-answering, and newly constructed Interleaved Image-Text (IIT) benchmarks. WINGS allows for maintaining text-only performance with limited resources and further enhances performance in well-resourced settings.

## Acknowledgments and Disclosure of Funding

This research was supported by National Science and Technology Major Project (2022ZD0114805), NSFC (61773198, 62376118,61921006), Collaborative Innovation Center of Novel Software Technology and Industrialization, CCF-Tencent Rhino-Bird Open Research Fund (RAGR20240101)

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

# Supplementary Material

## A Experimental Setups and Implementation Details

**Training Datasets.** The training datasets for the first and second stage of WINGS$_{\texttt{base}}$ are consistent with LLaVA$_{\text{v1.5}}$ [80]. For the second stage, WINGS$_{\texttt{pro}}$ extends the training dataset to include some visual QA datasets as ALLaVA [13], SynthDog [59], and ArXivQA [70], and text-only QA datasets as Stanford Alpaca [109], Alpaca GPT-4 [97], LIMA [146], UltraChat [31], WebQA [12], and BELLE-0.5M [53]. WINGS$_{1.8\text{B}}$ shares the same training set as WINGS$_{\texttt{pro}}$.

**Model Structures.** We employ Qwen1.5 [6] and SigLIP [140] as our foundations.

**Training Hyperparameters.** We utilize a batch size of 32, along with the AdamW optimizer and a cosine schedule. For all WINGS-series, the learning rate is set at $1e^{-3}$ for the first stage and adjusts to $2e^{-6}$ for the second stage, except for the projector as $1e^{-5}$.

**Training Environment.** WINGS$_{\texttt{base}}$ and WINGS$_{\texttt{pro}}$ are trained over approximately 1.5 or 6 days on $8\times$ A100 GPUs. WINGS$_{1.8\text{B}}$ require approximately 5 days of training on $4\times$ A100 GPUs.

## B Additional Experimental Results

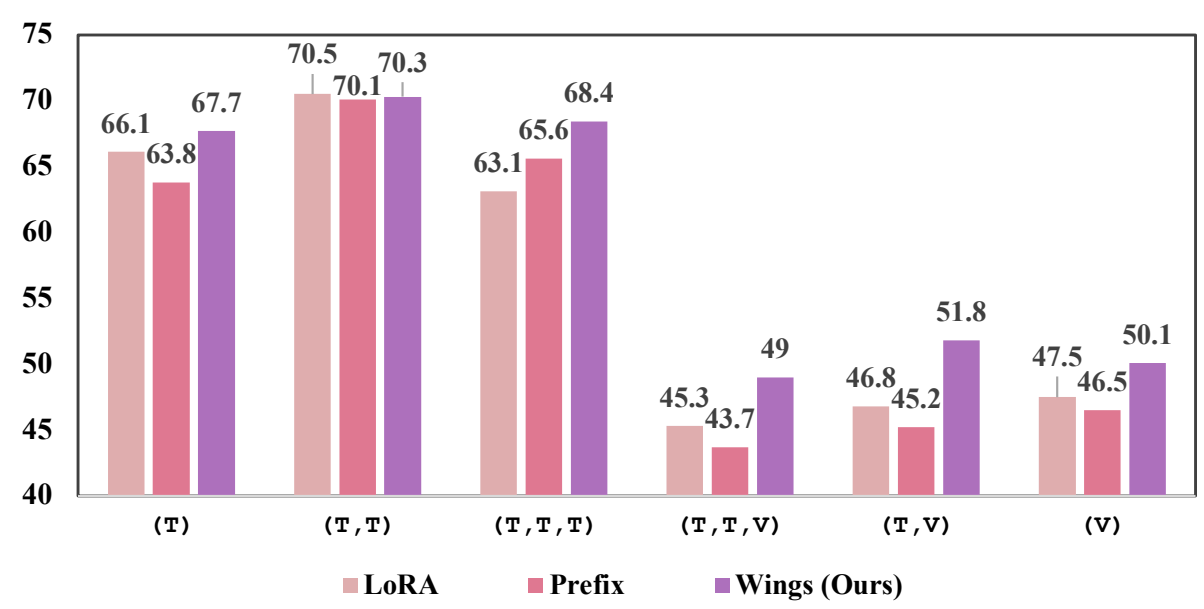

**(a)** Comparison of Parameter Efficient Modules and WINGS

Figure 6: **Performance comparison** on the newly constructed **Interleaved Image and Text (IIT) Benchmark** of the **Parameter Efficient Modules**. The horizontal axis represents different multimodal question settings. The horizontal axis shows different multimodal setups, *e.g.*, (T, T, I) represents a visual question after two text-only QAs.

Should we only add additional modules on top of an LLM branch or, like WINGS, create two distinct learners for visual and textual modalities? We delve into the low-rank adaptation (LoRA) [47] and Prefix-tuning [71] for minimally adapt to the LLM component. These techniques introduce optimization parameters beyond the primary branch. These lightweight adjustments align with extensive modifications, effectively minimizing text-only forgetting but concurrently curbing cross-modal positive transfer.

## C Discussion

WINGS is a universal plugin that can be integrated with any multimodal mixed-input MLLMs. Notably, it introduces a new concept of competitive reuse among multiple expert groups: we may not require the experts to the Transformer block's MLP layer at a scale three orders of magnitude larger; instead, a minor update in the attention for better allocation may suffice. This idea is also found in some variants of LoRA [36, 118]. In the future, we will gradually explore the future of MLLMs.

Regarding the extent to which WINGS alleviates attention shifts, we acknowledge that the main branch of WINGS (without visual and textual learners) still exhibits attention shifts. However, since the outputs from the visual and textual learners compensate within the hidden state, we extract portions of the attention weight matrices from both learners and add them to the main branch's weights. Results

show that throughout the training process, the attention shift phenomenon is continually mitigated under the influence of these learners. Essentially, the primary branch's attention ideally retains its original text-only capabilities, while the visual solutions are implemented within the auxiliary learner.

# D Limitation & Broader Impact

Despite WINGS' strong adaptability for embedding auxiliary attention learners in various MLLMs, integrating visual learners requires restarting the feature alignment training, incurring extra costs. Additionally, its deployment on edge devices faces limitations, with $\text{WINGS}_{1.8B}$ offering a solution at the expense of performance. Furthermore, WINGS still requires some text-only data to replay and enhance overall performance, aiming for integration into more generic AI systems in the future.

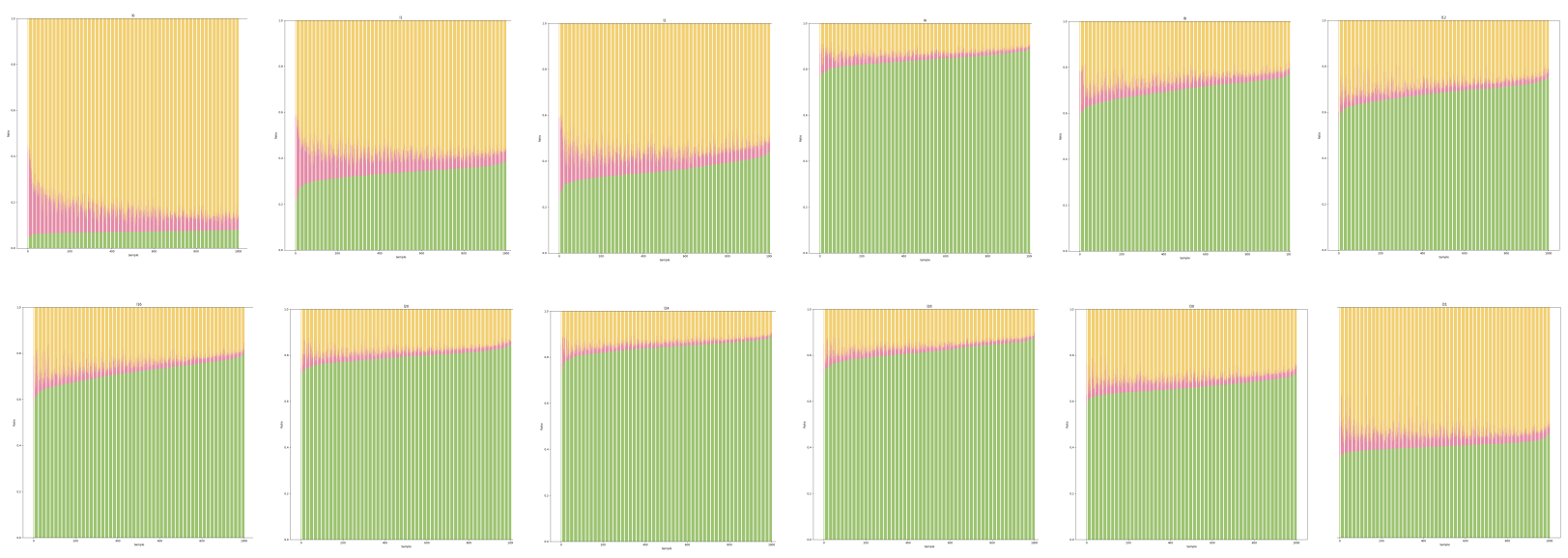

Figure 7: **Dynamics of Attention Weights from Shallow to Deep Layers.** We calculate the proportion of attention weights for the image-before (yellow), the image-itself (red), and the image-after (green) in each layer. From left to right, top to bottom, from shallow to deep layers.

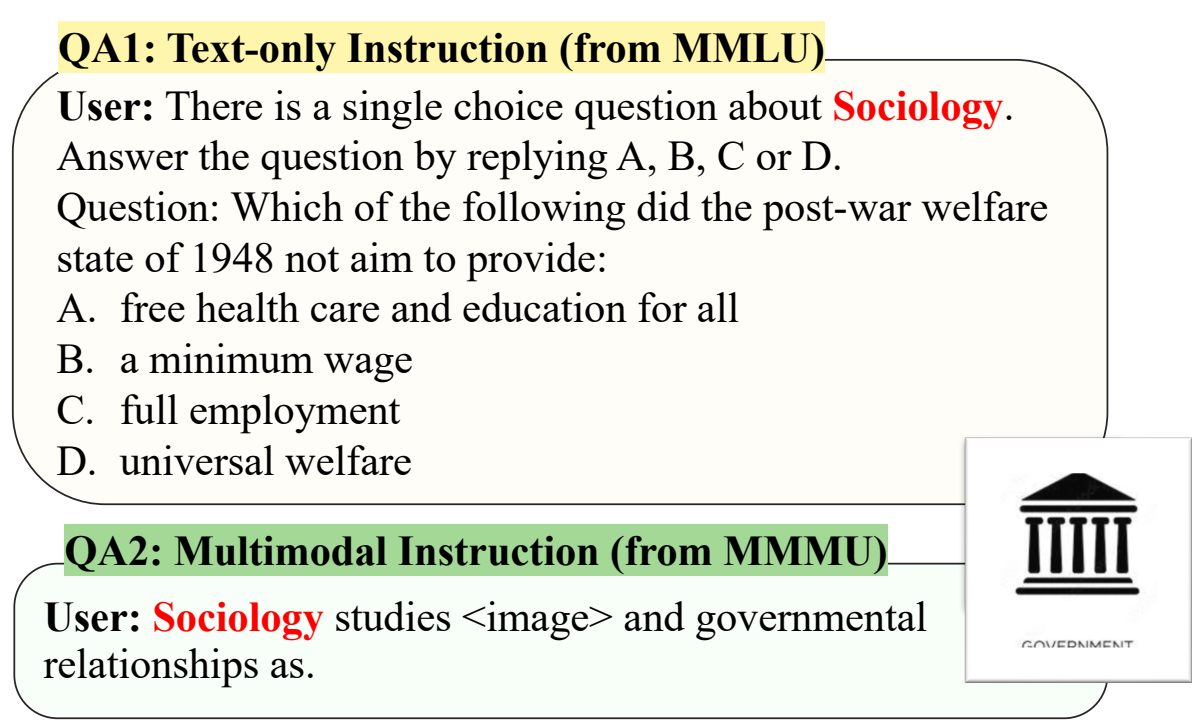

Figure 8: **An Example of an Interleaved Image-Text Benchmark.** This dialogue is represented as (T, V), consisting of a text-only QA from MMLU [43] and a visual QA from MMMU [138]. It can be observed that, due to the sampling, both include questions from the *Sociology* category.

