# OpenReview forum: "Wings: Learning Multimodal LLMs without Text-only Forgetting"
_NeurIPS.cc/2024/Conference — NeurIPS 2024 poster_

### Official Review · Reviewer_LUvX · 2024-07-08

**Soundness:** 3
**Presentation:** 3
**Contribution:** 3
**Rating:** 6
**Confidence:** 4

**Summary:**

This paper introduces the text-only forgetting phenomenon, where multimodal large language models (MLLMs) experience a significant decline in performance on text-only evaluations. The authors claim that this phenomenon is related to the attention shift of cross-layer MLLM-LAWS (Layer-level Attention Weights) before and after processing images. To address this, the authors propose WINGS, which utilizes visual and textual learners to compensate for the attention shift. Experiments across text-only, VQA, and IIT benchmarks demonstrate the effectiveness of WINGS.

**Strengths:**

1. The observation that text-only forgetting is related to MLLM-LAWS is intriguing.
2. WINGS shows significant improvements in text-only QA and multimodal QA tasks.
3. The paper is well-organized and easy to follow.

**Weaknesses:**

1. Although the paper provides a valuable observation about the correlation between text-only forgetting and MLLM-LAWS correlation in Figure 2, it lacks a deep discussion on the underlying reasons for this phenomenon. MLLM-LAWS correlation measures the attention states before and after the image, but in the text-only evaluation, no image involved. Thus, it remains unclear why text-only forgetting is related to MLLM-LAWS correlation.

2. Figure 2 shows that text-only performance is correlated with MLLM-LAWS. However, it would be interesting to know if the multimodal performance is also correlated with MLLM-LAWS. If so, the correlation score could be a valuable metric for MLLM model selection or evaluation.

3. Table 1 shows that LoRa efficiently mitigates the text-only forgetting problem but degrades multimodal performance. The authors do not provide any explanation for this observation.

**Questions:**

1. What are the 100 diverse MLLMs visualized in Figure 2(c)? Are these models sampled from the same type of MLLMs? Does the correlation occur in MLLMs trained on interleaved image-text datasets?

2. Is the relationship between text-only performance and MLLM-LAWS consistent across various scales of LLM backbones?

**Limitations:**

The major limitation of this paper is the lack of a deep understanding of the phenomenon shown in Figure 2. While the paper identifies a correlation between text-only forgetting and MLLM-LAWS, it does not explore the underlying mechanisms behind this relationship.

---

> ### Author Rebuttal · Authors · 2024-08-07
>
> Dear Reviewer LUvX,
>
> We sincerely thank Reviewer LUvX for the keen observations and suggestions, as well as the recognition of the attention shift and Wings effectiveness, and the overall flow of the writing. We will update all modifications in the final version. Thank you.
>
> * **Q1:** "MLLM-Laws w/ image *v.s.* text-only evaluation w/o image, why text-only forgetting is related to MLLM-Laws correlation"
>
> * **A1:** In Figure 2 of the main paper, the attention shift phenomenon guided by MLLM-Laws is shown to be strongly correlated with text-only forgetting, **following are the two main factors:**
>
>     * The attention shift in MLLM occurs because the model's LLM main branch **excessively focuses on image features**, causing a deviation in attention toward the images (especially post-image part, line 138). **This "over-focus" prevents effective information gathering in text-only scenarios**, leading to potential attention dispersion. We discussed this in details at the final version.
>
>     * Additionally, we have conducted supplementary experiments to support this hypothesis.
>
>     1. **In Figure 1 of the supplement PDF**, our analysis shows that a reduced attention shift results in more accurate and focused text-only attention distributions.
>
>     2. **In Table 1 of the PDF**, we propose a new metric for the variance of inter-word probabilities in text-only tasks, linking attention shift (MLLM-Laws induced) and text-only performance. **We find that attention shift can diminish MLLM's information aggregation capabilities in text-only data**; further experimental results will be published in the final version.
>
>     Furthermore, as described around line 231 in Figure 4(b) of the paper, a key step in compensating for attention shift is aligning the Learner with the main LLM branch during the initial training stage. We conducted additional experiments:
>
>     * **In Table 2 of the PDF**, we compared the Learners' attention weights after the first training stage with **the main branch's scale and the routing allocation weights after the second stage**. The Router's distribution weights indicate effective compensation for attention by the Learner structure.
>
>     * **In Table 3 of PDF**, comparing the training outcomes of Wings with baseline methods, we observed that Wings effectively mitigates attention shift.
>
>     * Finally, **In Figure 1 of PDF**, further analysis of the attention distribution in cases within Wings shows reduced shifts due to dominant segments.
>
>
> * **Q2:** "more analysis of attention shift *v.s.* multimodal performance"
>
> * **A2:** **In Table 11 of the PDF**, we found only a weak correlation in multimodal performance. For instance, on MMMU-VAL and ScienceQA, but it wasn't significant on MME.
>
>     Following LEEP and LogME's benchmark, **we developed a MLLM library for task-level model selection**, achieving improvements over random selection. Establishing a connection between attention shift and the generalized lower bound is essential for this process. Further analysis and results will be included in the final version.
>
> * **Q3:** "more explanation for LoRA observation"
>
> * **A3: LoRA reduces forgetting in text-only tasks but develops less multimodal capability.** Early LoRA studies often focused on simpler models and tasks, whereas multimodal tasks typically show high-rank weight variations. We demonstrate this **in Table 1 of the main paper under the same training data and network parameters.**
>
>     To verify that full parameter fine-tuning in multimodal scenarios **is not a low-rank perturbation, in Table 4 of the PDF**, we sampled weight transformations and performed singular value decomposition, **revealing significant changes in the weight matrix spectra from full fine-tuning**. It's important to note that our findings pertain to full fine-tuning, while Wings uses an independent module that interacts with cross-modal inputs. Therefore, the ranks identified do not directly indicate the rank needed in the Wings learner's LoRRA.
>
>     [1] LoRA Learns Less and Forgets Less.
>
> * **Q4:** "details of 100 MLLMs in Figure 2"
>
> * **A4:** In our training data, we adopted **various ratios of multimodal to text-only samples**: `25:1`, `20:1`, `10:1`, `5:1`, `2:1`, `1:1`, `1:2`, ..., `1:25`, along with an `all:0` (12 combinations, ensuring a sufficient amount of multimodal samples). We used learning rate 1e-3 for the first stage and 2e-6 or 1e-5 for the second. **We sampled 5 models in one epoch**, excluding 12 failed models due to issues like gradient explosion, resulting in 108 models for analysis.
>
>     Although these models share the same architecture (Qwen1.5 7B + SigLIP), performance varies significantly between text-only and multimodal scenarios. **The results reflect different training epochs, hyperparameters, and data, revealing potential correlations.** Nonetheless, they are classic and general, with a sufficient sample size.
>
> * **Q5:** "different scales of MLLMs for attention shifts"
>
> * **A5:** Generally, scaling laws suggest that model patterns can be transferred across different scales. **We conducted experiments using the 1.8B Qwen-Chat backbone**, training models with varying ratios of **text-only instruction data to multimodal data**: `10:1`, `2:1`, `1:1`, `1:2`, and `1:10`, alongside fully multimodal data for a total of six variations.
>
>     In the first training stage, we set the learning rate to 1e-3, adjusting it to 2e-6 in the second stage. We averaged 5 models per epoch, discarding 4 due to issues like gradient explosion, **resulting in 26 distinct models**. All models started with the same initialization and were trained on 4 A100 GPUs.
>
>     The results **in Table 12 of PDF** demonstrated that MLLM-Laws and text-only correlations were also evident in smaller-scale models.
>
> Please let us know if you have any further suggestions or questions! Thank you!
>
>
> Best regards,
>
> All Authors.

---

> > ### Comment · Reviewer_LUvX · 2024-08-11
> >
> > Thank the authors for the detailed response. The authors have addressed most of my concerns. I'd like to raise my rating to Weak Accept.

---

> > > ### Author Response · Authors · 2024-08-11
> > >
> > > Thank you for your updated rating! If you have any further questions or comments, please feel free to reach out.

---

### Official Review · Reviewer_TqCY · 2024-07-13

**Soundness:** 2
**Presentation:** 2
**Contribution:** 2
**Rating:** 5
**Confidence:** 5

**Summary:**

Multimodal large language models (MLLMs) are initiated with a trained vision encoder and LLM, then fine-tuned on multimodal mixed inputs. In this process, the LLM catastrophically forgets the text-only instructions. This paper first reveals that text-only forgetting is related to the attention shifts from pre-image to post-image texts. Based on this analysis, the authors proposed Wings, the complementary visual and textual learners. The experiment results show that Wings outperforms similar-scaled MLLMs in both text-only and visual question-answering tasks.

**Strengths:**

- The analysis of attention shifts is novel and interesting.
- The experiment results demonstrate the effectiveness of the proposed method in both text-only and multimodal benchmarks.

**Weaknesses:**

- The discrepancy seems to exist between the motivation/analysis and the method. There is a lack of detailed explanation on how the Wings module improves the correlation between pre-image and post-image attentions.
- The proposed Low-Rank Residual Attention (LoRRA) module appears to be a variant of LoRA, but there is no detailed motivation or comparative analysis provided.
- The ablation study is insufficient. Ablation experiments are only compared on the Interleaved Image and Text (IIT) benchmark. It would be better to include comparisons across various benchmarks used in the main results as well.
- The simple baseline to prevent forgetting is simply utilizing text-only data, yet there are no comparisons.

**Questions:**

- The most intuitive and direct approach to increase the attention correlation can be simply adding correlation loss between the attentions. What would the results be if we applied the correlation loss directly?
- The proposed method looks somewhat similar to P-LoRA [1]. How does it compare with this?
- For ablation studies (Figure 5), which model is used between Wings-base and Wings-pro?

[1] Dong, Xiaoyi, et al. "Internlm-xcomposer2: Mastering free-form text-image composition and comprehension in vision-language large model." arXiv preprint arXiv:2401.16420 (2024).

**Limitations:**

The paper addresses limitations in the paper.

---

> ### Author Rebuttal · Authors · 2024-08-07
>
> Dear Reviewer TqCY,
>
> We sincerely appreciate reviewer TqCY for acknowledging our analysis of attention shift and the effectiveness of Wings. All updates will be included in the final version. Thank you!
>
> * **Q1:** "how the Wings module improves the correlation"
>
> * **A1:**
>
>    * **From a structural perspective:** the Wings module operates **independently of the attention block**, with its outputs combined through weighted addition with the main branch LLM's attention to compensate for attention weight distribution.
>
>    * **In terms of the forward process:** the Wings module uses image features as keys and values, while the query derives from previous hidden states. This allows it to **focus more on the attention between the image and the pre-image and post-image**, compensating for the main branch LLM's attention weights.
>
>    * **From an optimization standpoint:** A key step in the Wings module's compensation for attention shift is aligning with the main branch **in the initial stage**. As shown in Figure 4(b) of the main paper, learners **first allocate attention between the pre-image, post-image, and image in the first stage**. Following this, the main branch LLM completes alignment under the "guidance" of the Wings module in the second stage, enhancing attention relevance for pre-images and post-images.
>
>    In the experiment described **in Figure 5 (C)**, we observe that compensation with the visual wings improves performance for both text-only and multimodal cases. We also conducted **additional experiments**:
>
>    * We hypothesize that models with significant pre-image and post-image attention shifts may have **a more dispersed** attention distribution for text-only tasks, with relevant analysis provided **in Table 1 of the supplemental PDF**.
>
>    * **In Table 2 of the PDF**, we compare the allocation weights of the Wings module and the main branch of the two stages. The weights from the router indicate that the Wings module **effectively compensates for pre-image and post-image attention** in the first stage.
>
>    * Additionally, **Figure 1 in the PDF** shows that the attention distribution **over words varies** with different models exhibiting attention shifts.
>
>
> * **Q2:** "detailed comparative analysis between LoRRA and LoRA" **and "compare to P-LoRA"**
>
> * **A2:** While the Low-Rank Residual Attention (LoRRA) structure also utilizes low-rank mappings, it fundamentally differs from Low-Rank Adaptation (LoRA) in key aspects:
>
>    1. **Motivation Design**: LoRRA functions **independently as an auxiliary module** to compensate for attention shifts in the main branch, while **LoRA** is **an efficient parameter tuning method** for adapting pre-trained models to target tasks with minimal training overhead.
>
>    2. **Structural Position and Training Strategy**: LoRRA operates as a separate module **that aligns prior to** attention distribution, whereas **LoRA and P-LoRA** are integrated within the attention module, **parallel to the internal attention linear mapping**, allowing the main branch mapping to remain unchanged while learning low-rank decomposition matrices. P-LoRA processes only forward image features on the low-rank mapping.
>
>    3. **Effectiveness**: Table 1 in our paper compares LoRA and Wings **under identical training data and base model architecture**. In multimodal scenarios, LoRA retains text-only content better but demonstrates weaker multimodal capabilities. This suggests its limitations in handling the complexity of multimodal tasks, as low-rank perturbations may be insufficient [1]. Results for P-LoRA with InternLM-XComposer2 7B can be found in Table 12 of the PDF.
>
>    [1] LoRA Learns Less and Forgets Less.
>
>
> * **Q3:** "ablation comparisons across various benchmarks in the main results"
>
> * **A3:**
>
>    * **Table 1 in the main paper** presents a comprehensive benchmark result from **16 text-only datasets** across **5 major domains**, including mathematics and coding, along with multimodal instructions (see Table 2, line 242). All methods in Table 1 used **the same training data and model architecture**, **serving as an ablation study**. The Wings architecture shows significant improvements over the baseline and LoRA with equal parameters and consistent training data.
>
>    * We added ablation experiments. **In Table 5 of the supplement PDF**, we compare ablation results across various benchmarks from the main results. Similar to IIT Bench, we found that **a lower learning rate or the inclusion of visual Wings** can maintain strong performance for text-only tasks (e.g., MMLU, WinoGrande).
>
> * **Q4:** "the baseline simply utilizing text-only (training) data"
>
> * **A4:**
>
>    1. Incorporating text-only training data during continuous training of MLLMs **helps prevent forgetting is common**, although it increases labeling and training costs. For instance, DeepSeek-VL used over 50% text-only data in its multimodal training.
>
>    2. The large parameter counts of LLMs complicate regularization and knowledge distillation, **making them less feasible**. Additionally, heightened expert supervision can be costly compared to collecting labeled text-only data.
>
>    3. While the MLLM community is evolving, some methods **lack open training data and may not align with publishers' hyperparameter settings**. Nonetheless, we've tested and compared most MLLMs in Tables 1 and 2 of the main paper.
>
>    4. In Figure 5 of the main paper, we introduced the Interleaved Image and Text (IIT) Bench for evaluations in continuous multimodal environments.
>
> * **Q5:** "add attention correlation loss".
>
> * **A5:** We attempted to impose KL divergence as a constraint, but this **made the training process unstable**. This is due to correlation **being a statistical phenomenon**, making it hard to control the scale of each sample. Results can be found in Table 12 (line 5) of the PDF.
>
> * **A6:** For ablation studies (in Figure 5 of the main paper), we used **Wings_base**.
>
> Best regards,
>
> All Authors.

---

> > ### Comment · Reviewer_TqCY · 2024-08-12
> > **Additional questions**
> >
> > First of all, thank you for your thorough response. After reading the response and the paper again, some of my questions were resolved, but I have a few more that I would like to ask:
> >
> > - Table 2 of PDF
> >   - Could you provide a detailed explanation of how the values in Table 2 were calculated? The caption mentions “distribution of weights between the learner and the main LLM branch,” but what specific weights are being referred to here? According to Eq (5), the router does not seem to assign weights to the LLM and each learner. How does this relate to the explanation?
> >   - Additionally, how do these weights in Table 2 indicate that the Wings module mitigates attention shifts?
> > - Role of the Router
> >   - What is the role of the router? According to Eq (5), the router does not control the weights between the LLM and visual/textual learners but rather adjusts the internal weights of the outputs of visual/textual learners. Why is this approach necessary, and what is the motivation behind it?
> >
> >
> > - Additional discussion
> >   - Wings introduce independent visual learners to prevent the LLM from over-relying on visual features. While this is explicit in terms of the model’s structure, it implicitly addresses the problem of over-dependence on visual features. In other words, even with this approach, there could still be an overall model tendency to depend too much on visual features. Could you discuss your thoughts on this?
> >   - Additionally, while the paper focused on attention shift and over-focusing on visual features, there might also exist a knowledge forgetting problem in the LLM. What are your thoughts on this?

---

> ### Author Response · Authors · 2024-08-12
> **Response to the Additional Questions - 1**
>
> Thank you for your detailed response!
>
> * **Part 1:** Table 2 of PDF
>    * How to calculate:
>        * In the first row of Table 2 in PDF, since the router is not introduced in the first stage, we calculate **the ratio of attention weights** between the visual learner and the LLM attention block **to reflect the importance weights.**
>
>            Specifically, we extract the above two sets of **attention weights** that image tokens receive (the shape is [sequence length x image token length]). We first average across the sequence dimension. Then, we calculate the ratio for each image token and average these ratios to obtain the results.
>
>            We sincerely apologize for the confusion. In our response to reviewer LUvX, we referred to "attention weights" instead of "weights."
>
>        * In the second row of Table 2, we show **the ratio between router weights** on the visual learner and the LLM attention block (set to 1) on image tokens.
>
>            As stated in Eq (5) in the main paper, the output of the LLM attention block (**with weights treated as 1**) is added to the output of the visual learner (with weights determined by the router), *e.g.*, `0.5681:1=0.3623:0.6377`.
>
>    * The explanation provided in Table 2 are:
>
>        1. **The first row** indicates that in the first stage, the visual learner **captures most of the attention** on image tokens due to the tuning strategy of Wings.
>
>        2. **The second row** shows that in the second stage, the router **assigns weights to the visual learner's outputs**, constraining its attention (as noted in Eq. (5) of the main paper). The weight ratio indicates the router prioritizes the main branch LLM's attention outputs, but it does not imply a mitigation of attention shift.
>
>        3. **In Table 3 of the PDF**, we calculate the number of samples where 'the correlation between MLLM-Laws Before Image and After Image' is positive, indicating **a smaller attention shift**. The Wings module mitigates attention shifts with more positive samples compared to LLaVA.
>
>       We appreciate your insights and hope those clarify the meaning of Table 2. **We sincerely apologize for any confusion.**
>
> * **Part 2:** Role of the Router
>    * **The role of the router:** The router's weight $r_a$ is multiplied with the outputs of the visual/textual learners and then added to the outputs of the LLM's attention block (as the Eq. (5)). This effectively establishes a routing ratio of $r_a$ : $1$ between the two components. The router can **adjust the compensation scale** of the visual/textual learners **for each token**.
>
>    * The router's motivation stems from the observation that **different tokens** demand **different levels of visual attention compensation [1]**. For example, when inquiring about spatial relationships, more visual cues are needed. In contrast, asking about the capital of the United States does not require visual information. Therefore, we need to customize the compensation of visual and textual attention for each token.
>
>       **[1]** Are We on the Right Way for Evaluating Large Vision-Language Models?
>
>    * We test the ratio between **router weights** on the visual learner and the LLM attention block in the perception and reasoning categories of MME data. The results show that in the perception category, which involves more image descriptions, the router weight **assigned to visual learners is higher**, and the image attention compensation is greater.
>
>    | Model | MME perception | MME reasoning |
>    |-------|----------------|---------------|
>    | Wings | 0.273:1 | 0.240:1 |
>
>    In summary, the router dynamically allocates the weights for visual/textual learners to compensate for attention to each token.
>
> The additional discussion will be presented in the following comments. Thank you!

---

> ### Author Response · Authors · 2024-08-12
> **Response to the Additional Questions - 2**
>
> * **Part 3.1:** Additional discussion -- Is the overall Wings over-relying on visual feature
>    * The overall Wings may not rely too much on visual features because the attention of **the textual learner** is focused on the text-only part. When there is too much reliance on visual features, the router will **assign a greater weight to the textual learner**. Wings essentially processes part of the visual feature attention through an "independent" module, learning how to allocate this along with the LLM attention block.
>
>    * Wings may have some limitations; for example, due to **the scarcity of spatial relationship instructions** in our training set at that time, the overall Wings might show the over-relying on rare spatial information. We are continuously improving this and will present it in the final version.
>
>    * It seems there may be an interest in expressing the concept of modality expansion: To prevent the LLM from over-relying on visual features, we propose Wings of LLM; if addressing **over-relying audio features** could pave the way for visual, textual, and audio Wings of LLM. That is a general and feasible exploration.
>
> * **Part 3.2:** Additional discussion -- Knowledge forgetting
>
>     * Knowledge forgetting **can be a part of** text-only forgetting. Wings also exhibit forgetting in specific areas, such as knowledge forgetting, such as an approximately 3% decline in the `logical_fallacies` category of the MMLU dataset. This may be related to the **easier-to-forget** reasoning knowledge.
>
>     * We also attempt to observe the relationship **between various metrics and forgetting rates** across different domains. For example, in the MathQA dataset, there is a correlation of approximately 0.629 (across five models) between attention shift and forgetting rates.
>
>     * Knowledge forgetting may relate to the **Incremental Learning**. Wings can add appropriate lightweight structures to achieve a better trade-off in the model's **stability-plasticity dilemma [2]**.
>
>     **[2]** DER: Dynamically Expandable Representation for Class Incremental Learning
>
> Thank you very much for your response. We sincerely look forward to your feedback. We are committed to continually improving Wings. Thank you once again!

---

> > ### Author Response · Authors · 2024-08-13
> >
> > We submitted our reply about eight hours ago, but it seems the system did not send a notification. When you have a moment, could you please take a look at our response? Thank you!

---

> ### Comment · Reviewer_TqCY · 2024-08-13
>
> Thank you for your response once again. However, I still have a few confusing points:
>
> - First, for clarity, let the router's output weights be $\mathbf w$, then, $\mathbf w \in \mathbb R^{s \times s}$ and it multiplies with learners' output of $\mathbb R^{s \times d}$. Each row is the output of a softmax, ensuring $\sum_j \mathbf w_{ij}=1$. Is this correct?
>   - Apart from this question, it would be beneficial to provide a clearer description of the router operation in the paper.
> - If so, it seems that the router’s role is to aggregate learner outputs on a token-wise basis. If the motivation is token-level scaling, defining $\mathbf{w}$ as a column vector ($\mathbf{w} \in \mathbb{R}^{1 \times s}$) might be more reasonable.
> - Additionally, I’m unsure whether the router’s design aligns with its intended motivation. To achieve token-wise compensation scaling, the router should adjust the balance between learners (i.e., assigning different weights to each learner) rather than focusing on intra-learner aggregation.
>
> If I have misunderstood anything, please let me know.

---

> ### Author Response · Authors · 2024-08-13
>
> Sorry, there may have been some misunderstandings. Let's clarify everything comprehensively:
>
> * **A1:** **Both the router's output and 'softmax'** are misunderstood.
>
>     $\mathbf{w}$ **are not** $\in \mathbb{R}^{s \times s}$. The router only activates the top-(sequence_length) columns of the single-layer MLP, meaning it maps **from the sequence dimension to one dimension**. Therefore, the router's output weights $\mathbf{w} \in \mathbb{R}^{1 \times s}$. There are **two** such single-layer MLPs **for both the visual learner and the textual learner.**
>
>     We denote the maximum value of the sequence length as $S$, the weight of the MLP (router) as $\mathbf{W} \in \mathbb{R}^{1 \times S}$, and the attention weights as $\mathbf{a}$. **We have**
>     $$
>     \mathbf{w} = \mathbf{W}[:, :s] \cdot \mathbf{a}^{\top}.
>     $$
>     Thus, $\mathbf{w} \in \mathbb{R}^{1 \times s}$. We guess that the misunderstanding comes from **the 'softmax' in line 193** of the main paper.
>
>     Exactly, the softmax is applied **between the weights of visual and textual learners**, **not** on the sequence_length_dim. *E.g.*, if **concat the $\mathbf{w}$ of visual and textual, and get a matrix** $\hat{\mathbf{w}} \in \mathbb{R}^{2 \times s}$,  we let $\sum_j \hat{\mathbf{w}}_{i j}=1$). As the visual/textual ratio in the first stage is 1:0, in the second stage, the 'softmax' ensures that the weights of both **are summarized to 1 for each token (like 1+0=1 in the first stage)**. We also mentioned related content in Reviewer GnKS's **A4**.
>
> * **A2:** After clarifying the misunderstanding from the previous question, this question **aligns with what you mentioned**; the router's output weights **are actually** $\in \mathbb{R}^{1 \times s}$.
>
> * **A3:** After explaining the previous misunderstanding, it can be observed that the balance between visual and textual learners is achieved through the router's 'softmax' (at the token level). This ensures **the sum of weights for each token is related and their gradients are interconnected**. Furthermore, the router weights differ and are correlated across visual and textual learners, varying for each token while **preserving a balance among the visual and textual learners.**
>
> **We sincerely apologize** for the misunderstanding. **The motivations you understand and the router structure you find reasonable are indeed what we designed in Wings.** We apologize again for our insufficient expression. **We will make sure to provide detailed explanations in the final version.** We’re truly sorry for the inconvenience this has caused you.
>
> Thank you very much! If you have any more questions, please feel free to let us know.
>
> **Thank you once again!**
>
> We provide **the code for the Router as follows:**
> ```
> import torch
> import torch.nn as nn
> import torch.nn.functional as F
> from transformers.activations import ACT2FN
> from typing import Optional
>
> class Router(nn.Module):
>     def __init__(self, max_length: int, act_fn: Optional[str] = 'silu'):
>         """Router class.
>
>         Args:
>             max_length (int): The maximum value of the sequence length.
>             act_fn (Optional[str]): The activate function.
>         """
>         super().__init__()
>         self.linear = nn.Linear(max_length, 2)
>         self.act_fn = ACT2FN[act_fn]
>
>     def forward(self, self_attn_weights, **kwargs):
>         """forward of Router.
>
>         Args:
>             self_attn_weights (torch.Tensor): The attention weight from the LLM (main branch).
>                 shape: [batch_size x num_heads x sequence_len x sequence_len]
>
>         Returns:
>             list of [torch.Tensor, torch.Tensor]: Two Tensors, router weights on visual and textual learner for each token, respectively.
>                 shape: [[batch_size x sequence_len x 1], [batch_size x sequence_len x 1]]
>         """
>         cur_weights = torch.sum(self_attn_weights, dim=1) # sum on head_dim
>
>         cur_weights = self.act_fn(F.linear(
>             input=cur_weights,
>             weight=self.linear.weight[:, :cur_weights.shape[-1]], # split to sequence_length, cur_weights.shape[-1] is sequence_length
>             bias=self.linear.bias
>         )) # route with MLP
>         cur_weights = cur_weights.softmax(dim=-1) # softmax on two learners
>         cur_weights = torch.split(cur_weights, split_size_or_sections=1, dim=-1)
>         return list(cur_weights)
>
> batch_size, num_heads, sequence_length = 2, 32, 5
> print(f'Start testing, batch_size: {batch_size}, num_heads: {num_heads}, sequence_length: {sequence_length}')
>
> router = Router(max_length=2048)
> self_attn_weights = F.softmax(torch.rand((batch_size, num_heads, sequence_length, sequence_length)), dim=-1) # attention weights
> router_weights = router(self_attn_weights) # forward
>
> print(f'The router\'s output weights of the visual learner, shape: {router_weights[0].shape}') # [2, 5, 1]
> print(router_weights[0])
> print(f'The router\'s output weights of the textual learner, shape: {router_weights[1].shape}') # [2, 5, 1]
> print(router_weights[1])
> ```

---

> > ### Comment · Reviewer_TqCY · 2024-08-14
> >
> > Thank you for the clarification. Then, it seems that Eq (5) is incorrect. It would be beneficial to update the equation correctly.
> >
> > I appreciate your proactive and sincere engagement during the discussion period. To conclude, I would like to summarize my review:
> >
> > - As I noted in my initial review, I believe the text-only forgetting problem targeted by Wings is very important, and the performance is impressive.
> > - During rebuttal and discussion, several concerns are resolved, but major concerns remain:
> > 	- The attention shift analysis is interesting but not fully convincing in terms of generalizability.
> > 		- Although the analysis involves over 100 MLLMs, the models are trained by adjusting the ratio between multimodal and text-only data within the same dataset and sampling models at different training steps in a single run. This raises doubts about whether the findings would apply to MLLMs with entirely different data distributions, optimization strategies, or architectures.
> > 	- It is also unconvincing that Wings effectively resolves attention shifts, as it does not directly address the problem. There is no explicit regularization to ensure the suppression of attention shifts.
> > 		- Even though the Wings module operates independently from the main LLM attention blocks, its attention output is still combined with the main block, which means it can still overly rely on visual features. The authors describe Wings' role as "compensation," but if the compensation scale is too large, it essentially becomes another form of attention shift.
> >
> > For these reasons, I believe this paper remains borderline. However, given the importance of the problem and the impressive performance, I am inclined to raise my score to borderline accept. If there's anything I may have missed or misunderstood regarding my remaining concerns, please feel free to correct me and share your thoughts.

---

> > > ### Author Response · Authors · 2024-08-14
> > >
> > > Thank you very much for your response and support!
> > >
> > > * The study of Attention Shift in the 100 MLLMs does include some overlapping training epochs. However, their training data also contains certain differences (as adjusting the ratio of multimodal to text-only data **requires random sampling**). Recently, we have been researching how to generate better training data for MLLMs. In the final version, **we will train more MLLMs to supplement additional experiments as much as possible**. Furthermore, we will consider different optimization strategies, alignment methods, architectures, and model performances to investigate the generalizability of attention shifts further. Thank you very much for your comprehensive and detailed suggestions.
> > >
> > > * Wings is a better structural and strategic assistant for LLMs to enhance learning from visual input. The overall intuition behind Wings is that the existing LLM structure **is prone to attention shift when new image (visual modality) inputs**. In the first stage, the visual wings **concentrate on the main visual attention** (during which the compensation scale may be too large for certain samples, as you mentioned). However, in the second phase, through the weight allocation of the router and the learning of the textual wings, attention **is balanced between the LLM branch and the visual wings**. This balancing relationship in the learning process resembles **a form of regularization**: for example, when attention toward images increases, the textual wings **also drive an increased focus on the text-only parts**. In Wings, not all inputs will pass through both sides of the learner (for instance, text-only data will not pass through the visual learner), so we prefer to view Wings as **a suite of structural regularizations** that help LLMs learn better during multimodal extension.
> > >
> > >     Given the differentiation of tokens, we cannot achieve an explicit regularization constraint here because we lack prior information that applies universally to each token; not all tokens should necessarily focus less on images.
> > >
> > >     We are attempting to extend Wings to more modalities. Thank you very much for your effective suggestions, and we will continue to focus on **improvements related to regularization constraints**.
> > >
> > > **Thank you once again!** We've learned a great deal from our discussion with you. **We will certainly continue to work hard to develop the more generalized MLLM. We truly appreciate your support! Thank you!**

---

### Official Review · Reviewer_3jFi · 2024-07-13

**Soundness:** 3
**Presentation:** 3
**Contribution:** 2
**Rating:** 6
**Confidence:** 4

**Summary:**

This paper addresses a significant challenge that when Multimodal Large Language Models (MLLMs) as they expand LLMs’ capabilities to include vision tasks. Specifically, it highlights the issue of "text-only forgetting," which occurs when MLLMs trained with visual inputs struggle to effectively process text-only instructions. The problem is attributed to inconsistent attention shifts between text and visual inputs before and after training. To solve this, this paper proposes Wings, a method adding visual and textual learners to balance attention shifts. Their approach shows improved performance on both text-only and multimodal tasks, including a newly constructed Interleaved Image-Text (IIT) benchmark for mixed-modality interactions.

**Strengths:**

1. The writing in the article is clear.
2. Noticing the text-only ability of MLLMs for a more general model is important, this ability can be lost when turning LLMs into MLLMs.
3. Existing methods need a lot of text-only training samples, which limits improving MLLM performance. The proposed model shows good results.

**Weaknesses:**

The paper provides a detailed analysis of the impact of image tokens on the distribution of existing tokens and introduces the Wings structure. However, to ensure that the Wings structure effectively minimizes the overhead of text-only training data while maintaining model performance, more time-based quantitative analyses are necessary to demonstrate the module's effectiveness.
It is observed that methods like DeepSeek [1], which have published their performance on text-only datasets, do not show significant performance degradation. This might be due to their extensive use of text-only training data. Thus, there is a trade-off between the overhead of text-only training data (i.e., excessive training costs) and model optimization strategies (or training methods). Please show why Wings is efficient in this trade-off. Specifically, to what extent can text-only data be reduced in Wings training process while still maintaining text-only performance?
In Figure 2, it is shown that image tokens are inserted among text tokens. Is this insertion specific to chat models with system prompts, or does it also occur in base-version models?
Concerning the IIT benchmark, when constructing multimodal samples (e.g., (T,T,V) as shown in Figure 5), how are these samples constructed, and how is dataset balance maintained? Please provide additional implementation details.

[1] DeepSeek-VL: Towards Real-World Vision-Language Understanding.

**Questions:**

1. Wings note that the text-only performance of MLLMs and the performance of the initial LLMs. This makes a compelling argument for a more general model. Could you provide examples illustrating the phenomenon of performance degradation on text-only in existing models?
2. In line 131, how is the activation value for each token computed for each layer? Is the averaging done along the columns or rows of the attention matrix, i.e., which dimension is averaged?
3. There are other methods for adding additional modules to MLLMs, such as CogVLM [1]. Please clarify the similarities and differences, particularly why Wings-architecture can alleviate the forgetting problem in text-only data.
4. Provide more specific implementation details of the model structure mentioned in line 161?
5. Please include more ablation studies on training Wings on the base LLM, such as the effect when the initial LLM is base-version and does not have a system prompt. If the images are always positioned at the beginning, does Wings still work effectively?
6. Please provide more experimental results regarding the efficiency gains with LoRRA. Compared to LoRA, it specifically handles modalities. How does its efficiency compare to LoRA, and what are its advantages?
7. Please provide more inference details, particularly how the data flows through the Wings modules (the learners for each modality) and the routing mechanism.
8. In Figure 4, the second stage requires fine-tuning the LLM parts of the MLLM. How much resource overhead does this require? Is it significantly more than methods like LLaVA?
9. Please provide the detailed parameter counts for the 1.8B and 7B model versions and include detailed descriptions of each dataset in subsequent versions.
10. In the IIT benchmark, why does the (T, T, T) configuration not show performance improvement compared to (T, T)? Does it indicate the presence of noisy text-only few-shot samples?

Some Tips:
Please add a description of $\mathbb{1}$ and its subscript in Eq. 3.
Add details to the weighted sum description in line 189, specifically how the attention weight matrix is applied in the router module.
The ARC-Easy dataset mentioned in line 217 is missing a citation.
What is referred to by "Efficient Multimodal LLMs" in line 260?
Provide a more detailed explanation of Partitioned LR in Figure 5.
Please include the prompts used to generate data with GPT-3.5 Turbo mentioned in line 268 in subsequent versions.

[1] CogVLM: Visual Expert for Pretrained Language Models.

**Limitations:**

The Limitation section in the appendix states that a relatively small amount of text-only training data is still needed to activate the MLLM's capabilities, which is reasonable. Additionally, Wings is a multimodal framework trained from a general LLM. The generalization capabilities and other limitations of Wings can be comprehensively evaluated in the future.

---

> ### Author Rebuttal · Authors · 2024-08-07
>
> Dear Reviewer 3jFi,
>
> Thank you very much for Reviewer 3jFi's detailed feedback. We appreciate the reviewer's recognition of Wings' motivation, writing, and overall performance. All updates will be incorporated into the final version. Below are our responses to the clear comments and questions raised:
>
> * **Q1:** "trade-off between training cost and performance"
>
> * **A1:** Wings shows **a direct improvement in training efficiency**: when adding an equivalent amount of Text-only data to the training set, Wings achieves better Text-only performance and improved multimodal performance while maintaining a training cost nearly identical to the baseline LLaVA. To achieve comparable performance, **the baseline requires more Text-only training data.** Table 1 in the main paper uses entirely consistent training data, demonstrating that Wings can maximize overall general performance within a balanced trade-off. Additionally, **we compared the same-architecture baseline using Wings_pro training data with Wings_base** (which utilized less training data, as elaborated in line 675 of the paper). We found that Wings can deliver stronger overall Text-only and multimodal capabilities with a smaller dataset.
>
> * **Q2:** "Wings for LLM_base"
>
> * **A2:** Wings proves effective for LLM_base as well: the presence of image tokens in any position of the training data aids the model in understanding the context of images and text. The placement of image tokens **does not restrict** Wings' expression and **does not affect** the attention weight extraction in Wings' LLM main branch. Results from training on the Qwen1.5-7B base LLM are presented **in Table 13 of PDF**, showing sustained excellent performance.
>
> * **Q3:** "details of Interleaved Image and Text (IIT) Bench"
>
> * **A3:** As discussed in line 264 of the paper, we first sampled multimodal examples **from MMMU, MMBench, SEED-Bench, and AI2D, then collected semantically similar text-only samples from MMLU, CMMLU, OpenbookQA, and HellaSwag.** For some questions, **we used GPT-3.5 Turbo for polishing**. We performed clustering to sample different semantic clusters, aiming for a diverse dataset, and we will open-source the entire dataset after the paper is accepted.
>
> * **Q4:** "case of text-only forgetting"
>
> * **A4:** In Table 2 of the main paper, we present comparisons with other methods, such as Qwen-VL and the LLaVA series models. **These models did not specifically integrate text-only data into their training or address the issue of text-only forgetting**, which led them to underperform compared to the initial text-only LLM (as shown in Table 2 of the main paper).
>
> * **Q5:** "router details (with attention inputs)"
>
> * **A5:** We begin by performing linear mapping on the softmax dimension. Below, we detail how the Wings structure extracts attention weights for routing, collaborating with the learner for inference:
>
>    1. First, we obtain the LLM attention weights from the main branch: during inference, by setting `output_attentions=True`, **the attention weights are derived by multiplying the query and key, dividing by the square root of head_dim, and applying softmax** (note that dropout is not applied). Consequently, the shape of the attention weights is [`number of heads` x `sequence length` x `sequence length`], ignoring the batch size dimension, and the sum along the last dimension equals 1.
>
>    2. For the image tokens, Wings introduces a visual learner. **The router maps the attention weights to a weight ratio between the image learner and the main LLM**: first, we sum the weights along the head dimension and then apply an MLP (linear mapping followed by an activation layer) to produce a shape of [`sequence length` x `2`], assigning weights to each token. The visual learners for text tokens will be masked, but in the second stage, textual learners will also undergo training in a similar manner. **This weighted combination forms a new set of hidden states, which are then processed in the next layer.** We will provide a detailed description and pseudocode in the final submission.
>
> * **Q6:** "learner of Wings *v.s.* CogVLM visual expert"
>
> * **A6:**
>    1. **Positioning differs**: The visual expert in CogVLM is a parallel module through linear mapping, while the LoRRA in Wings operates independently from the entire attention block.
>    2. **The motivation is similar but slightly distinct**: Similar to Mix of Experts and P-tuning, visual experts enhance the attention module's ability to **learn cross-modal interactions and alignments**. Wings also boosts multimodal learning capabilities, specifically addressing compensatory attention for text-only forgetting.
>    3. **Utility varies**: Visual experts increase training overhead significantly and, without low-rank optimization, **will substantially raise network parameters** (though inference overhead remains nearly unchanged). In contrast, Wings adds only a minimal number of parameters (**refer to Table 9 of PDF**).
>
> * **Q7:** "details on line 161"
>
> * **A7:** Line 161 refers to a visual encoder, specifically the `vit_so400m_patch14_siglip_384` and a linear mapping with a single-layer MLP.
>
> * **Q8:** "more studies on Wings with base-version LLM"
>
> * **A8:** **In Table 12 of the PDF**, we evaluate the MLLM based on the Qwen-base model (the image is at the beginning of instructions and lacks a system prompt). **The results show good performance**; however, because the base model has weaker instruction capabilities, **a standard chat LLM is generally more suitable**. We will include related experiments in the final version.
>
> Due to space constraints, **subsequent responses will be presented in the comments**.
>
> Best regards,
>
> All Authors.

---

> > ### Comment · Reviewer_3jFi · 2024-08-10
> >
> > Thank you for the detailed response. The authors have successfully addressed nearly all of my initial concerns, and based on the clarifications provided, I am inclined to raise my score.
> >
> > After considering the feedback from other reviewers, particularly the points raised by reviewers TqCY and LUvX, I am curious about how Wings compensates for attention shift. Your explanations concerning the structure, forward mechanism, optimization, and experiments are convincing.
> >
> > However, I have one additional question based on your explanation of the optimization process. You mention that "learners are first trained, so allocate attention in the first stage." Given that much of the data in the first stage is caption instruction, could you please elaborate on the nature of the visual attention that compensates for the LLM branch? Understanding this aspect more clearly would help in fully appreciating the robustness of your approach.

---

> > > ### Author Response · Authors · 2024-08-10
> > >
> > > Thank you for your feedback; it inspires our continuous improvement.
> > >
> > > To address your question about the nature of visual attention in the first training stage, we utilize caption data to align visual information. The Wings module enhances visual attention **related to image descriptions** while improving the model's **perceptual capabilities**. Unlike other MLLM baseline methods that only consider the visual part at the input layer, the Wings module **integrates images and captions across all layers**, deepening the LLM branch's understanding of visual content. However, further reasoning abilities still require the LLM main branch **to learn from instruction training data in the second stage**.
> > >
> > > For instance, after the first stage of training, Wings can interpret images and manage tasks like identifying "Is the word in the logo 'the beatles story liverpool?'" within the `perception OCR` category in the MME dataset. However, its reasoning abilities are less robust.
> > >
> > > The table below compares the performance of **baseline and Wings in `perception` and `reasoning` on the MME dataset**. It shows that Wings outperforms baseline methods in perception after the first training stage, with reasoning abilities improving further in the second stage.
> > >
> > > | Model | MME perception | MME reasoning |
> > > |-------|----------------|---------------|
> > > | LLaVA (after the first stage of training)  | 1197.53 | 216.18 |
> > > | Wings (after the first stage of training)  | 1286.46 | 241.80 |
> > > | Wings (after the second stage of training) | 1411.76 | 342.07 |
> > >
> > > Thank you once again for your support and recognition!

---

> > > > ### Comment · Reviewer_3jFi · 2024-08-12
> > > >
> > > > Thank you for your detailed response. I find the comparison of perception and reasoning results on MME across different stages convincing. I'm also curious if the improvement in reasoning after the second stage is due to the textual module branch of Wings, as I notice the textual wings fine-tuning was introduced in the second stage. If possible, could you provide additional information or experimental results on this? Thank you.

---

> > > > > ### Author Response · Authors · 2024-08-12
> > > > >
> > > > > Please wait. We have retrained the second stage on more text-only data. A complete response will be provided in about 7 hours. Thank you!

---

> > > > > > ### Author Response · Authors · 2024-08-13
> > > > > >
> > > > > > Thank you sincerely for your response!
> > > > > >
> > > > > > In comparison to visual learners, textual learners focus more on the attention between **the text-only part** and the sequence, compensating for the shifts in attention at each token (as mentioned in line 149 of the main paper). When the text-only part significantly influences the responses, the textual learner may **compensate with more attention**. To verify this, we extract images and instructions from **the MMMU-VAL dataset** in the categories of `business_economics`, `business_marketing`, `health & medicine_basic_medical_science`, `science_biology`, and `science_chemistry`. Additionally, we extract text-only instructions from **the MMLU dataset** corresponding to the categories of `econometrics`, `marketing`, `professional_medicine`, `college_biology`, and `college_chemistry`.
> > > > > >
> > > > > > We then **combine the images with** their strongly related instructions (from MMMU-VAL) and their weakly related ones (from MMLU), *e.g.*, image with **`business_economics` of MMMU-VAL** and the same image with **`econometrics` of MMLU**. The inference results on Wings show that for instructions from MMLU (text-only is more important), the routing weight assigned to the textual learner is greater. Furthermore, in Table 5 of PDF, we compare the situation without textual learners in the 'Only Visual Wings', revealing a decline **in both text-only and multimodal performance**. We also included experiments **with more text-only training samples**, which resulted in improvements in the text-only parts.
> > > > > >
> > > > > > |  | MMLU | ARC-Challenge | WinoGrande | MMMU-VAL | ScienceQA |
> > > > > > |-------|------|---------------|------------|----------|-----------|
> > > > > > | Wings | 60.5 | 43.4 | 69.3 | 39.9 | 78.8 |
> > > > > > | w/ more text-only training data | 60.7 | 44.8 | 70.1 | 39.9 | 79.2 |
> > > > > >
> > > > > > Thank you again for the discussion. We will continue to work hard to improve Wings! Thank you once again!

---

> > > > > > > ### Comment · Reviewer_3jFi · 2024-08-13
> > > > > > >
> > > > > > > Thank you for your additional explanations and the detailed results provided in the rebuttal. I find your clarifications convincing and I am glad to raise my score. I would also recommend including the results and explanations provided during the rebuttal period in the final version of the paper.

---

> > > > > > > > ### Author Response · Authors · 2024-08-13
> > > > > > > >
> > > > > > > > We are very pleased and grateful for your feedback. We will work on refining the final version to make Wings even better. Thank you very much!

---

> ### Author Response · Authors · 2024-08-07
>
> **We add the remaining responses here. Thank you!**
>
> * **Q9:** "more inference details"
>
> * **A9:** Starting from Figure 4 in the paper, we observe that the Learners' structure is embedded at each layer. The visual features from the first layer are fed into the key and value, **while the mapping of query, key, and value is implemented using a residual low-rank mapping**. We modified the fully connected layers in the attention module to use residual and low-rank mapping. In the output mapping matrix, we removed the residual connection since we initialized W_a in the low-rank mapping with Random Gaussian and W_b with Zero. **This allows the structure parallel to the attention module to start training with zero additions**, facilitating quicker adaptation to multimodal scenarios.
>
> * **Q10:** "details of training cost"
>
> * **A10:** **In Table 9 of the PDF**, we present network parameters and costs (in TFLOPS), revealing only **an increase of about 0.1B in parameters and 0.2 TFLOPS in cost**. The training time on 8 A100 GPUs increased by less than 1.5 hours.
>
> * **Q11:** "1.8B and 7B parameters count"
>
> * **A11:** Refer to **Table 9 in the PDF** for parameter counts and forward costs. We commit to providing dataset details in the final version.
>
> * **Q12:** "(T, T, T) weak but (T, T) strong in Figure 5"
>
> * **A12:** We appreciate the reviewer’s keen observation. This may be due to the two text-only in-context samples **belonging to two significantly different datasets**. We re-sampled (T, T, T) **to belong to the same dataset**, and the accuracy of Wings on (T, T, T) improved from 68.4% to 69.7%.

---

### Official Review · Reviewer_GnKS · 2024-07-15

**Soundness:** 3
**Presentation:** 2
**Contribution:** 3
**Rating:** 6
**Confidence:** 4

**Summary:**

The paper addresses how to solve the well-known problem of multimodal large language models (MLLMs), text-only forgetting referring to the phenomenon of MLLMs showing drastic performance drops on text-only instructions. The paper first observes based on the analysis over 100 MLLMs that the performance drop is related to attention shifts in text tokens before and after visual tokens within mixed visual-and-textual inputs. To compensate for the attention shift, the paper adds separate visual and textual learners with a router to each of the LLM blocks. The visual and textual learners are implemented with a newly designed Lor-Rank Residual Attention (LoRRA). The proposed method, Wings, achieves improvements over baselines on text-only benchmarks while achieving competitive or better performance on multimodal benchmarks, including a newly collected interleaved Image-Text (IIT) benchmark.

**Strengths:**

- The overall presentation of the paper is quite good. The analysis that explores over 100 MLLMs is really impressive. Especially, the Figure 2 effectively shows the observation that is the key to the proposed method, Wings.
- The proposed visual and textual learners make sense based on the observation.
- Wings achieves superior performance on extensive text-only and multimodal benchmarks.

**Weaknesses:**

This work is really intriguing. However, the rationale behind how the authors designed the proposed Low-Rank Residual Attention (LoRRA) as well as the ablation studies proving some critical design choices are missing.
For example,
- why did the authors add the residual terms (the identity matrices in Equation 4)?
- please provide the results without the router, if possible.

Also, please provide computation overheads of Wings in FLOPs, and how much does Wings increase training and inference time (comparison between Wings_base and the Qwen + SigLIP model)?

**Questions:**

- Please provide the details of 100 MLLMs explored for the attention shift analysis.
- How can we get a, the attention weights of shape sxs from the LLM main branch? And please elaborate more on how the router operates on a, that is, more details about computing weighted sums of visual and textual learner outputs.
- In Figure 5 (b), did the authors ablate learning rates in the second stage of training? in other words, did the authors use the same learning rate of 1e-3 in the first training stage?
- Does the authors plan to release the second-stage training data for Wings_pro, and the new IIT benchmark?

Minor comment: If the indicator variable becomes zero for any i in equation (3), the probability is zero. Is the equation (3) correct, or there are some typos?

**Limitations:**

The paper addressed the limitations of the work.

---

> ### Author Rebuttal · Authors · 2024-08-07
>
> Dear Reviewer GnKS,
>
> We appreciate Reviewer GnKS's thoughtful feedback and support, especially regarding the motivation, structure, and performance of our Wings. In response to these valuable insights, we have conducted additional experiments and enriched our descriptions to reinforce our approach. All modifications will be highlighted in the final version.
>
> * **Q1:** "Wings design rationale and ablation studies",
>    * **Q1.1**: the residual terms.
>    * **Q1.2**: w/o the router.
>
> * **A1:** The primary rationale for Low-Rank Residual Attention (LoRRA) is to mitigate the attention shift seen in MLLM when integrating visual inputs (as noted in lines 147-149). The main features include:
>
>    * **Lightweight, Independent Attention Module:** LoRRA improves multimodal attention by adding **an independent learner module** that manages visual interactions with minimal parameters. It generates keys and values from initial visual features and queries from previous hidden states. This reduces the visual attention load on the primary branch, allowing for more focused processing of text parts.
>
>    * **Pre-alignment Process in the First Stage:** In the initial training stage, LoRRA is learned **on inter-modal adaptation using the learner's visual attention modules**, without the router's involvement.
>
>    * Regarding the ablation study of structural design: the construction of the LoRRA structure underwent several unreliable iterations. **In PDF Table 6**, We conducted **ablation studies focusing on the attention component**, as training MLP mappings is resource-intensive. Our results indicate that incorporating **linear mappings, learnable prompts, or dynamic mixed LoRA structures** can improve multimodal perception. However, these methods often suffer from limited learning, resulting in a trade-off where less forgetting leads to diminished learning outcomes.
>
>    **Responses to Q1.1 and Q1.2:**
>
>    * **A1.1: Table 7 in PDF** demonstrates that the **text-only input performs slightly worse than the LoRRA structure with residuals**, which enhances gradient propagation and promotes faster learning. The residual connections create a more coherent overall structure by simplifying the fully connected mappings to an identity mapping.
>
>    * **A1.2: In Table 8 of PDF**, the results indicate that random allocation (w/o a router) significantly disrupts attention and impairs performance, while 1:1 allocation reduces the effects of scale transformation. **As noted in lines 281-283 of the main paper**, the router weights are also crucial for multimodal capabilities.
>
> * **Q2:** "details of computation overheads"
>
> * **A2: Table 9 of PDF** summarizes the computational overheads of Wings in (Tera) FLOPS, **alongside parameter counts and TFLOPS costs compared to the Qwen + SigLIP baseline**, using measurements from the [`calculate-flops.pytorch`](https://github.com/sovrasov/flops-counter.pytorch) library with a batch size of 1 and a maximum sequence length of 2048.
>
> * **Q3:** "details of 100 MLLMs for the attention shift analysis."
>
> * **A3:** We analyzed **108 models derived from** 12 multimodal training configurations (`25:1`, `20:1`, `10:1`, `5:1`, `2:1`, `1:1`, `1:2`, etc., up to `1:25` and `all:0`) with varying multimodal:text-only data ratios and 2 learning rates (1e-3 for the first, and either 2e-6 or 1e-5 for the second stage), assessing their MMLU, 5-shot performance differences in text-only and multimodal contexts, **as illustrated in Figure 2 of the main paper**.
>
> * **Q4:** "details of attention weights"
>
> * **A4:** Wings leverages attention weights from transformers by **enabling access during inference through `output_attentions=True`**. The attention weights are generated by performing a query-key-value operation, where the query from the LLM's main attention module multiplies with the key, is divided by the square root of the head dimension, and **is processed with a softmax function—without any dropout**. These attention weights, **shape [`number of heads`, `sequence length`, `sequence length`], are summed over the last dimension to yield values that equal 1**. For image tokens, wings uses a visual learner where the router maps attention weights by first summing along the head dimension and then **applying an MLP for transformation into a two-dimensional output**. Consequently, this reshapes the attention weights to [`sequence length` x `2`], allowing the router to assign specific attention weights to each image token.
>
> In the second stage, we incorporate **textual learners alongside masked visual learners to create weighted hidden states** from both image and textual tokens, with a detailed process description and pseudocode provided in the final version.
>
> * **Q5:** "learning rate of the second stage"
>
> * **A5:** We used a learning rate of 1e-3 for the first training stage and 2e-6 for the second, while the visual projectors’ alignment modules were set to 1e-5. **Our comprehensive ablation**, as detailed **in Table 10 of PDF**, tested learning rates of 2e-6, 6e-6, and 1e-5, applying larger fine-tuning steps for the visual component to **adapt better to multimodal inputs** and **reduce text-only forgetting**, similar to models like Qwen-VL and DeepSeek-VL.
>
> * **A6:** We are committed to **releasing all data, model weights, and code, along with the new Interleaved Image and Text (IIT) benchmark**.
>
> * **Q7:** "i in equation (3)"
>
> * **A7:** In the main paper, we adapted equation 3 from LLaVA to account for interleaved image tokens within the text, omitting loss computation for the "next image token" **since the MLLM cannot generate image tokens.** This adjustment results in a text-only indicator variable interval of `[1, v_start)` U `(v_end, s]`. We will elaborate on this point in detail in the final version.
>
> Thank you once again for your constructive feedback, which will greatly enhance the clarity of our paper.
>
> Best regards,
>
> All Authors.

---

> > ### Comment · Reviewer_GnKS · 2024-08-10
> >
> > Thank the authors for the detailed response. The authors addressed most of my concerns as well as those of other reviewers. I will keep my rating.

---

> > > ### Author Response · Authors · 2024-08-11
> > >
> > > Thank you very much!

---

### Author Rebuttal · Authors · 2024-08-07

Dear Reviewers,

**Thank you for your meticulous observations and analyses.** We are thrilled that you have recognized our work, Wings. We appreciate your acknowledgment of Wings’ **effectiveness on** text-only, multimodal, and Interleaved Image and Text (IIT) benchmarks. We are particularly pleased that you found the concept of **attention shift** intriguing and that you described the paper as clear and easy to read.

We observed **the attention shift phenomenon** in MLLMs that have forgotten their text-only capabilities. To address this issue, we designed Wings with **the Low Rank Residual Attention (LoRRA) structure**, which significantly **enhances performance without** adding high costs. During this time, we've been continuously improving Wings to achieve even better results.

Once again, thanks to each reviewer and everyone involved! We have responded to all your questions in detail and **included additional tables and figures in the PDF**. **Should you have further inquiries, please feel free to reach out. We appreciate your support and will continue to work diligently to enhance Wings.**

Best regards,

All Authors.

---

### Decision · Program_Chairs · 2024-09-25

**Decision:**

Accept (poster)

**Comment:**

This paper unanimously receives positive rates thanks to clear representation and convincing experiments. Although overall reviews are positive, the additional description in the rebuttal should be reflected in the final draft.